# Activity dependent feedback inhibition may maintain head direction signals in mouse presubiculum

Jean Simonnet[1,†], Mérie Nassar[1,2], Federico Stella[3], Ivan Cohen[4], Bertrand Mathon[1], Charlotte N. Boccara[3], Richard Miles[1] & Desdemona Fricker[1,2,†]

Orientation in space is represented in specialized brain circuits. Persistent head direction signals are transmitted from anterior thalamus to the presubiculum, but the identity of the presubicular target neurons, their connectivity and function in local microcircuits are unknown. Here, we examine how thalamic afferents recruit presubicular principal neurons and Martinotti interneurons, and the ensuing synaptic interactions between these cells. Pyramidal neuron activation of Martinotti cells in superficial layers is strongly facilitating such that high-frequency head directional stimulation efficiently unmutes synaptic excitation. Martinotti-cell feedback plays a dual role: precisely timed spikes may not inhibit the firing of in-tune head direction cells, while exerting lateral inhibition. Autonomous attractor dynamics emerge from a modelled network implementing wiring motifs and timing sensitive synaptic interactions in the pyramidal—Martinotti-cell feedback loop. This inhibitory microcircuit is therefore tuned to refine and maintain head direction information in the presubiculum.

[1] Inserm U1127, CNRS UMR7225, Sorbonne Universités, UPMC Univ Paris 6 UMR S1127, Institut du Cerveau et de la Moelle Épinière, Paris 75013, France. [2] CNRS UMR 8119, Université Paris Descartes, Paris 75006, France. [3] Institut of Science and Technology Austria, Klosterneuburg 3400, Austria. [4] INSERM U1130, CNRS UMR8246, Sorbonne Universités, UPMC Univ Paris 6 UM CR 18, Neuroscience Paris Seine, Paris 75005, France. † Present addresses: Université Paris Descartes, CNRS UMR 8119, 45 rue des St-Pères, 75006 Paris, France (D.F.); Bernstein Center for Computational Neuroscience, Humboldt University of Berlin, Philippstrasse 13, Haus 6, 10115 Berlin, Germany (J.S.). Correspondence and requests for materials should be addressed to J.S. (email: jean.simonnet@bccn-berlin.de) or to D.F. (email: desdemona.fricker@parisdescartes.fr).

Head direction signals are relayed by the thalamus[1] and processed in several interconnected brain regions. They include the presubicular cortex, located between the hippocampus and the entorhinal cortex[2]. About half of presubicular principal neurons signal head direction[3,4]. They fire persistently when the head of the animal faces a specific direction. The dorsal presubiculum, also termed postsubiculum (Brodmann area 48), controls the accuracy of head direction signals and links them to specific features of the environment. Presubicular circuits are thus central to the role of the hippocampal formation in landmark-based navigation[5,6].

Vestibular inputs make a decisive contribution to head directional firing of neurons in the anterodorsal nucleus of the thalamus[1,7,8] and lesions of this thalamic region abolish head direction firing in presubiculum[6]. Head direction signals transmitted via the thalamus are integrated in the presubiculum with visual information[5] from visual[9] and retrosplenial cortices[7], and information from the hippocampal formation[2]. Presubicular head direction cells in layer 3 project to the entorhinal cortex[10,11] and may contribute to spatial firing of grid cells[12–14].

The properties of presubicular microcircuits that signal head direction are less clear than the long-range outputs from the region. The electrophysiological and morphological properties of excitatory and inhibitory presubicular neurons have been described[15,16]. Pyramidal cells (PC) can generate persistent firing with little adaptation over tens of seconds[17] as needed to signal a maintained head direction. However, less is known of the connectivity and dynamics of inter- and intralaminar presubicular synapses[18]. Such data are crucial to understand how signals are transformed within the presubiculum and how this structure gates the flow of head direction information to the entorhinal cortex.

The roles of presubicular interneurons are presumably multiple: they provide global inhibition to restrain over-excitation[19] and, as suggested by continuous attractor theories, could induce selective inhibition of PC, ensuring head direction signal specificity over time[14,20–23]. Yet, details of the recruitment of inhibitory cells are unknown. In somatosensory cortex, high-frequency PC firing is needed to recruit Martinotti interneurons. These cells then initiate a feedback inhibition of distal PC dendrites[24–26], to exert a local control on excitatory synapses made at these sites[27]. Facilitating excitation of interneurons may be critical for the treatment of the persistent head direction signal, however, there is no data on the functional effects of Martinotti cells (MC) in the presubiculum.

We report here that strong recurrent connectivity between the presubicular MC and layer III PC form a feedback inhibitory circuit. Importantly, the excitation of MC by PC exhibits a dramatic activity-dependent facilitation. The feedback effects of Martinotti-cell inhibition on pyramidal cell activity depend on the timing of the inhibitory post-synaptic potential (IPSP), suggesting they could provide a source of lateral inhibition that enforces directionally selective firing. Testing these hypotheses by modelling connectivity and synaptic dynamics of recurrent Martinotti cell-mediated inhibition revealed features of an attractor network generating activity patterns comparable to presubicular recordings *in vivo*. Our results demonstrate autonomous dynamic activity in the presubicular cortex emerging from the local circuits that process head direction signals *in vivo*.

## Results

**Electrophysiology of presubicular PC and MC.** We first characterized the basic properties of MC and PC in microcircuits of superficial layer 3 of mouse presubiculum. Data was obtained from 166 PCs and 161 MC recorded in horizontal slices (Fig. 1) from 60 animals. MCs were identified as GFP positive neurons in tissue from X98-SST and Sst-Cre::tdTomato transgenic mice[16].

MC often discharged spontaneously from membrane potentials depolarized above $-60\,\mathrm{mV}$ ($n = 80$ cells; Fig. 1a–c, Supplementary Table 1; ref. 16). Current pulses elicited low threshold spiking. The axons of these cells ramified extensively in layer 1 like MC in somatosensory cortex[28] (Fig. 1a,d,e). PC typically did not discharge spontaneously and membrane potentials were more hyperpolarized, below $-70\,\mathrm{mV}$ ($n = 87$ cells), than those of MCs (Mann–Whitney test, two-tailed $P < 0.0001$; Fig. 1a–c and Supplementary Table 1). PCs fired regularly in response to injected current[15] with a higher threshold current ($92.3 \pm 50.4$ pA, mean $\pm$ s.d.; $n = 65$) than in MCs ($51.5 \pm 38.9$ pA, mean $\pm$ s.d.; $n = 64$; Mann–Whitney test, two-tailed, $P < 0.0001$). The input–output gain was lower in PCs ($0.373 \pm 0.127$ Hz pA$^{-1}$, mean $\pm$ s.d.; $n = 65$) than in MCs ($0.845 \pm 0.040$ Hz pA$^{-1}$, mean $\pm$ s.d.; $n = 64$; Mann–Whitney test, two-tailed, $P < 0.0001$, Fig. 1e and Supplementary Table 1).

**Thalamic fibres directly excite principal neurons of layer 3.** Head directional inputs to the presubiculum originate in part from the anterior thalamic nuclei[6,29] (ATN). We sought to define presubicular targets of these afferents by *in vivo* stereotaxic, intra-thalamic injection of viral vectors to transduce channelrhodopsin-2 fused to eYFP (Fig. 2; $n = 5$ SstCre::tdTomato mice). Fluorescent (eYFP)-labelled thalamic axons innervated superficial layers of presubiculum, more densely in layers 1 and 3 than layer 2. A few axons projected to deep layers and parasubiculum. Thalamic axonal projections terminated abruptly at the border with the subiculum. Few, if any, axons innervated entorhinal cortex (Fig. 2a,b; Supplementary Fig. 1). Optical stimulation of ATN axons *in vitro* let us compare responses of PCs and MCs to thalamic input (Fig. 2c,d). At $-65\,\mathrm{mV}$, 11 out of 14 layer 3 PC were made to fire by optical stimulation, while identical stimuli induced firing in 4 out of 9 layer 3 Martinotti-like cells. The latencies of optically evoked excitatory postsynaptic currents (EPSCs) in PC were short and monosynaptic (median $= 1.435$ ms) and median charge transfer over 25 ms was 2.818 nC. The latencies of optically evoked EPSCs in Martinotti-like neurons were longer median $= 3.864$ ms with lower charge transfer (median $= 0.200$ nC over 25 ms) indicating a weaker excitatory drive (Fig. 2c,e,f). Tetrodotoxin (TTX) (1 μM) and 4-aminopyridine (4-AP) (100 μM) let us examine synaptic excitation directly mediated by thalamic afferents. Optical stimulation continued to excite PCs (median $= 1.306$ nC over 25 ms, Fig. 2d,g), but light-evoked responses in MCs were suppressed (median $= 0.013$ nC over 25 ms, Fig. 2d,g). These data suggest that optical excitation of MC is mediated indirectly via synapses made by presubicular PC. MC could provide a recurrent inhibitory control of PC. We next examined presubicular PC–MC connections and their dynamic behaviour in dual patch clamp records in slices produced from X98-SST mice.

**PCs excite MCs and receive recurrent feedback inhibition.** PC and MC were highly interconnected (Fig. 3) as expected from the spatial overlap of their axons and dendrites (Fig. 1a). The proportion of connected pairs was 57% (83 of 146 tested) for Martinotti cell to pyramidal cell (MC-to-PC) and 38% (59 of 156 tested) for pyramidal cell to Martinotti cell (PC-to-MC). 28% of cell pairs (39 of 141) were reciprocally connected, a little more than the 22% expected given the probability for unilateral connections. For PCs that excited a MC, the probability of reciprocal inhibitory connection was 81% (39 out of 48 tested). Only 48% (39 of 80 tested) of MCs inhibiting a PC received reciprocal excitation. Connectivity between pyramidal neurons (PC-to-PC) was very low (1 of 48 tested). At $-50\,\mathrm{mV}$, the mean amplitude of inhibitory postsynaptic currents (IPSCs) was

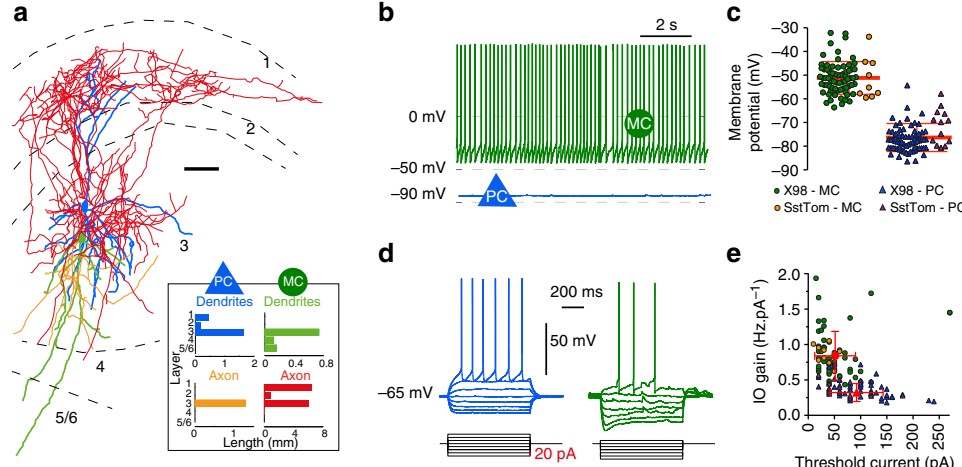

**Figure 1 | Morphology and electrophysiological properties of Martinotti interneurons and PC in layer 3 of presubiculum.** (**a**) Anatomical reconstruction of reciprocally connected PC and MC in layer 3. PC dendrites (blue) and MC axon (red) covered layer 1 and 3, PC axon (yellow) and MC dendrites (green) mainly occupied layer 3 (inset). Subiculum is to the left and the parasubiculum to the right. Scale bar, 50 μm. (**b**) Current clamp recordings of a MC (green) and a PC (blue). The MC fired spontaneously, the PC, with a more hyperpolarized membrane potential, was silent. (**c**) Membrane potential values for 80 MCs (circles, X98-SST, green; SstCre, orange) and 87 PCs (triangles, X98-SST, blue; SstCre, purple). The horizontal bar indicates the mean value, error bars represent s.d. (**d**) Typical responses of a MC and a PC to negative and positive current step injections of duration 800 ms from −65 mV. (**e**) Plotting input–output (I–O) gain against threshold current separates PCs (triangles) from MCs (circles; same colour code as in **c**). For detailed statistics see Supplementary Table 1.

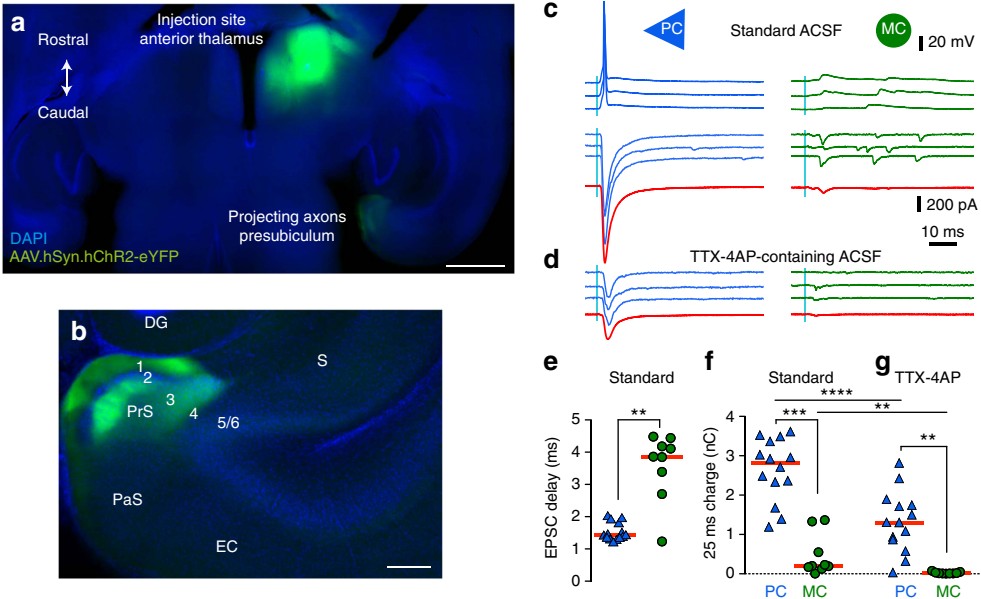

**Figure 2 | PC but not MC are directly innervated by the ATN.** (**a**) Channelrhodopsin-2-eYFP expression in ATN of SstCre::tdTomato mice 15 days after stereotaxic injections of an AAV vector. Injection site in AT and projecting thalamic axons in the ipsilateral presubiculum (horizontal brain section, 20° angle; dorso-ventral depth 2.5–3). Scale bar, 1 mm. (**b**) Enlarged view of the parahippocampal cortex. Thalamic axons specifically target the superficial layers of presubiculum. Scale bar, 200 μm. DG: dentate gyrus; S: subiculum; PrS: presubiculum, PaS: parasubiculum; EC: entorhinal cortex; (**c**) Presubicular responses to illumination (470 nm LED, 0.5 ms, 2 mW) of ChR2-expressing thalamic fibres, in simultaneous records from a PC and MC. Top, action potentials were evoked in three trials following a blue light flash in the PC but not in the MC. Below, light-evoked EPSCs from three trials, and average responses (40 trials) in red. (**d**) In TTX-4AP containing ACSF, EPSCs were still elicited in PCs, indicating that thalamic axons made direct synaptic contacts onto PCs. Responses were mostly abolished in MCs, suggesting indirect, disynaptic excitation of MCs. (**e**) Onset latencies of PC EPSCs are significantly shorter than for MC EPSCs. **Mann–Whitney $U$ test , $P < 0.01$. (**f,g**) Absolute charge transfer over 25 ms after the light stimulus in **f** standard and **g** TTX-4AP containing ACSF was significantly higher in PCs than MCs. *** and **Dunn's multiple comparison test $P < 0.001$ and $P < 0.01$, respectively, performed after a significant ($P < 0.0001$) Kruskal–Wallis test comparing responses in PC and MC from standard and TTX-4AP conditions. For PCs, the charge transfer was significantly reduced (****Wilcoxon-signed rank test $P < 0.0001$) but still present in TTX/4AP condition. For MC cells, the measured charge transfer was also significantly reduced (**Wilcoxon-signed rank test $P = 0.0078$) to values close to zero in TTX/4AP condition. Horizontal bars indicate median values. Data from $n = 14$ PC (blue triangles) and 9 MCs (green circle) from 5 mice.

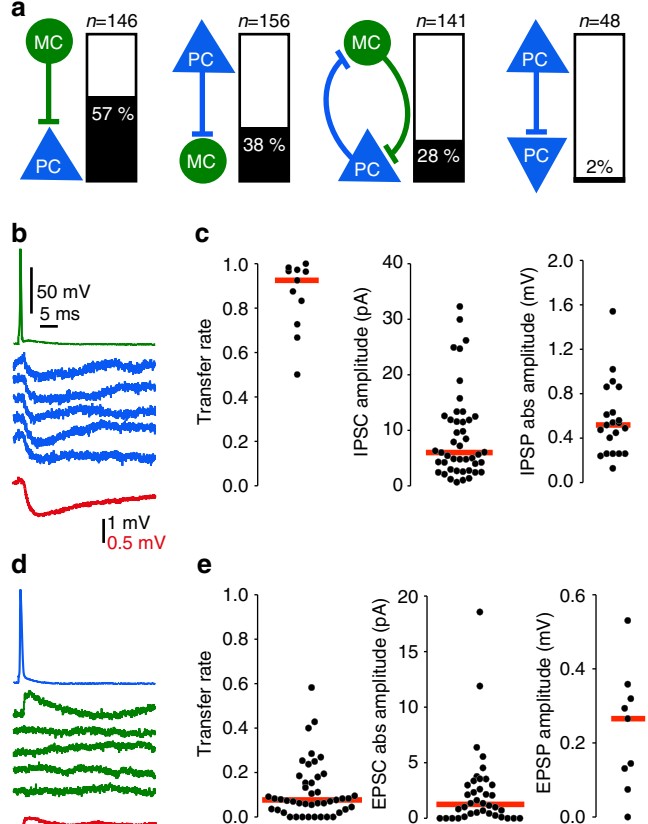

**Figure 3 | Pyramidal and MC form an inhibitory feedback loop.**
(**a**) Connectivity between PC and MCs, determined from dual patch clamp recordings in 55 X98-SST mice, showing the percentage of connected pairs (%) and the number of tested paired records (n). Very few PC-PC connections were detected. (**b**) Single spikes of MCs (green) reliably evoked small IPSPs in a PC (blue). Action potentials were initiated by brief current injections (1–2.5 nA for 1–2.5 ms (**b,d**)). Average current traces in red, stimulation artifacts on the MC voltage trace blanked. (**c**) Transfer rate (n = 11), IPSC amplitude (n = 45 pairs) and IPSP absolute amplitude (n = 21 pairs) from responses to single spikes at MC-to-PC synapses. Red horizontal bars are median values. Transfer rates from automatically detected synaptic events, recorded with a Cs-Glu based internal solution. Amplitudes from averages of responses recorded using a low-Cl K-Glu internal solution (cf. Methods). (**d**) Single spikes of PCs (blue) did not reliably evoke EPSPs in a MC (green). The mean EPSP amplitude (red) was very low. (**e**) Transfer rate (n = 44), EPSC absolute amplitude (n = 38) and EPSP amplitude (n = 8) from responses to single spikes at a PC-to-MC synapse. Red horizontal bars give median values. Transfer rate and potency from automatically detected synaptic events, recorded with a low-Cl K-Glu internal solution (methods). Efficacy was calculated as transfer rate × potency and used as measure of average amplitude.

9.01 ± 1.19 pA (n = 45) and in current clamp, IPSP amplitude was -0.56 ± 0.07 mV (n = 21). Single Martinotti cell spikes triggered inhibitory events with high probability (transfer rate 0.86 ± 0.05; median = 0.925; n = 11; Fig. 3c,d), with at least one postsynaptic event observed for each connected pair. PC-to-MC transmission was much less reliable. The single spikes transfer rate from pyramidal cell to Martinotti cell was 0.12 ± 0.02 (median = 0.08; n = 44, Fig. 3e). In six pairs, transmission was detected only during high-frequency trains; single presynaptic spikes evoked no excitatory post-synaptic potential (EPSP) in at least 30 trials for each pair. In 38 PC-to-MC pairs, single pyramidal cell spikes, or first spikes in a train, occasionally initiated excitatory

postsynaptic responses. Their potency, the mean absolute amplitude of single successful responses for PC-to-MC synapses, was 20.1 ± 1.94 pA (median = 20.4 pA; n = 31) or 1.44 ± 0.21 mV (median = 1.37 mV; n = 8). The efficacy, the potency multiplied by the transfer rate, was 2.36 ± 0.58 pA (median = 1.24; n = 38) or 0.24 ± 0.05 mV (median = 0.27; n = 9) for the first spike; Fig. 3e; Supplementary Fig. 2).

These data reveal an asymmetric synaptic reliability in recurrent pathways involving PC and Martinotti interneurons in superficial layers of the presubiculum. Inhibitory synapses are much more reliable than excitatory connections. Since the dynamic behaviour of both synapses in this feedback circuit governs its operation[30,31], postsynaptic responses at different rates of presynaptic firing were examined. Synaptic transfer rate, potency and efficacy were analysed for responses to trains of 30 action potentials at 10 or 30 Hz. The interval between tests was at least 20 s (Figs 4a and 5a). All postsynaptic events were detected and monosynaptic events were identified as described in methods.

**Stable Martinotti-cell inhibition during repetitive stimulation.** Information transfer at MC-to-PC connections was reliable and stable during activation at 10 or 30 Hz (Supplementary Table 2; Fig. 4a–d). Synaptic efficacy for the first five action potentials (early efficacy) and the last five action potentials (late efficacy) of trains of 30 presynaptic spikes at 10 and 30 Hz were similar (early 10 Hz, 16.97 ± 3.58 pA; late 10 Hz, 17.33 ± 3.32 pA; early 30 Hz, 16.13 ± 3.73 pA; late 30 Hz, 15.47 ± 2.76 pA, n = 8 pairs; Friedman test, P = 0.5222). Changes in efficacy during repetitive firing resulted from changes in potency rather than transfer rate (Fig. 4e). Cumulative efficacy evolved linearly during repetitive stimulation (Fig. 4f). Changes in synaptic frequency (see Methods) were proportional to changes in presynaptic firing frequency (Fig. 4g). Thus, MC-to-PC inhibitory synapses have stable dynamic behaviours, rather independent of previous presynaptic firing.

**Frequency-dependent unmuting of the PC-to-MC connection.** In contrast, PC-to-MC excitatory synapses displayed remarkable facilitating dynamic behaviour (n = 58/59 pairs). Figure 5 shows an example of EPSCs elicited by stimulations at 10 and 30 Hz (Fig. 5a). The low initial synaptic efficacy increased greatly with both the number and frequency of presynaptic action potentials (Supplementary Table 3; Fig. 5a–d, Friedman test, P = 0.0002). At 10 Hz, late efficacy (6.51 ± 2.99 pA, n = 9) was more than double early efficacy (2.99 ± 0.99 pA, n = 9). At 30 Hz, late efficacy (17.34 ± 5.14 pA, n = 9) was four times higher than early efficacy (4.41 ± 0.88 pA, n = 9). Efficacy increased for 6/9 pairs tested at 10 Hz (Fig. 5d left, Dunn's multiple comparison, n.s.) and for 9/9 pairs at 30 Hz (Fig. 5d right, Dunn's multiple comparison, P < 0.05). The extent of the increase varied between connections especially at 30 Hz (Fig. 5d). In contrast, efficacy changes at the PC-to-MC synapse during 30 Hz activation resulted from changes in response probability rather than potency (Fig. 5e, n = 15 pairs). Increased efficacy implies PC-to-MC synapses transmit more reliably as numbers and frequencies of presynaptic spikes increase. The synaptic frequency increased supra-linearly with presynaptic spike frequency. After one second, cumulative efficacy was 11 times higher at 30 Hz than at 10 Hz (Fig. 5f). The synaptic frequency was 5.9 times faster for early spikes and 7.9 times faster for late spikes, when presynaptic firing rate increased from 10 to 30 Hz (Fig. 5f,g), exceeding changes expected for a threefold increase in the rate of synaptic activation. We refer to this extreme form of facilitation as synaptic unmuting.

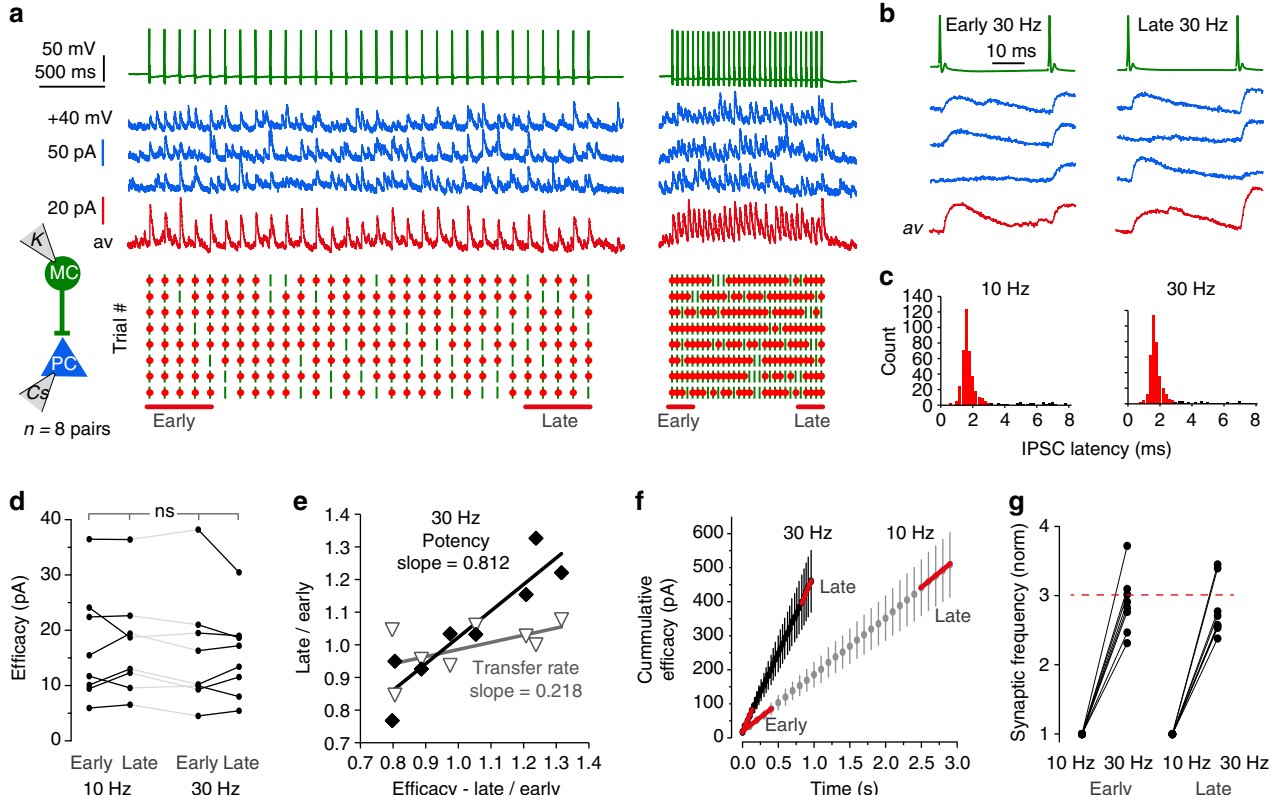

**Figure 4 | Stable inhibitory control by MC.** (**a**) 30 action potentials elicited at 10 (left) or 30 Hz (right) in a MC (green traces). Voltage clamp traces from a connected PC (blue traces) recorded at +40 mV, with a Cs-Glu internal solution. Three successive trials in blue, average of eight trials in red. The inter-trial-interval here was 40 s. Below, raster plots of synaptic transfer for eight successive trials, showing presynaptic action potentials (green bar) and IPSCs (red dots) triggered at monosynaptic latencies (0–3 ms). Transmission failed only infrequently (cf. Supplementary Table 2). (**b**) Detail of early and late MC spikes and IPSCs in 30 Hz trains. Stimulus artifacts blanked. (**c**) Poststimulus-histograms of IPSCs at monosynaptic latencies in the range 0–3 ms show peaks at 1.63 for trains at 10 Hz and 1.67 ms for trains at 30 Hz. (**d**) MC-to-PC synaptic efficacy (transfer rate x absolute potency) was unrelated to the spike position in a train or to firing frequency ($n = 8$, Friedman test, $P = 0.5222$). (**e**) Late/early transfer rate and potency plotted against late/early efficacy ($n = 8$ pairs, 30 Hz stimulation). These synapses are highly reliable with a low dynamic range (0.8–1.4). Slope of linear regressions show a small variation in synaptic efficacy is related to change in potency, rather than transfer rate. (**f**) The cumulative efficacy (mean ± s.e.m., summed efficacy over time) of MC-to-PC synapses reveals stable dynamics during long stimulus trains. (**g**) Synaptic frequency for early and late spikes of 10 and 30 Hz trains, normalized to 10 Hz. The increase in synaptic frequency corresponds to the threefold increase in presynaptic spike frequency (red dashed line).

**Lasting increase of transfer rate at the PC-to-MC synapse.** Presubicular MC are reliably excited only when PC fire at high frequency, as when they signal a preferred head direction. Our data showed synaptic efficacy returned to previous values within ∼ 20 s (Fig. 5a).

However, even if frequency-dependent changes are not maintained, medium-term effects on transmission at PC-to-MC synapses were detected ($n = 10$, Fig. 6; Supplementary Fig. 3). When synapses were unmuted by high-frequency (30–40 Hz) firing of the pyramidal neuron, transfer at the PC-to-MC synapse remained enhanced, as PC firing returned to lower frequencies (Supplementary Fig. 3a–c, blue lines). Similarly, after the PC-to-MC synapse was unmuted by high-frequency firing, a subsequent stimuli at 10 Hz maintained transfer rates well above those during a 10 Hz control spike train (Supplementary Fig. 3d,e). These results show that the unmuting effect of high-frequency synaptic activation outlasts the high-frequency stimulation itself. We determined the time course of decay of synaptic enhancement. Following synaptic unmuting by a 30 Hz spike train of duration 2 s, synaptic responses to test stimuli at 2 Hz revealed a double exponential decay of synaptic efficacy and transfer back to baseline level, with a fast time constant of 0.74 s and 0.97 s followed by a slower time constant of 12.96 s

and 18.94 s ($n = 7$ pairs, Fig. 6a–c). We conclude that the PC-to-MC synapse remained in a mode of enhanced information transfer efficiency for tens of seconds after sustained activation at high frequency.

**How head direction signalling activates the PC–MC loop.** How do these findings relate to head direction signalling *in vivo*? We recorded firing of presubicular head direction (HD) cells from rats in an open field in order to test their effects at the PC–MC synapse. Firing of presubicular head direction cells *in vivo* was very irregular and instantaneous frequencies fluctuated widely[3]. Neurons with typical mean direction specific firing rates of ∼ 15 Hz, could have peak instantaneous firing frequencies up to 250 Hz ($n = 5$; Fig. 7a,b). While the head remained within preferred directions of 200°–240°, the neuron shown in Fig. 7a fired in a sustained manner.

We used spike trains from isolated single head direction units within their preferred range *in vivo* as depolarizing current commands to presynaptic PCs in paired PC–MC recordings (Fig. 7c,d). At the start of high-frequency *in vivo* spike patterns, excitatory transmission was poor. The PC-to-MC synaptic efficacy increased considerably during sustained

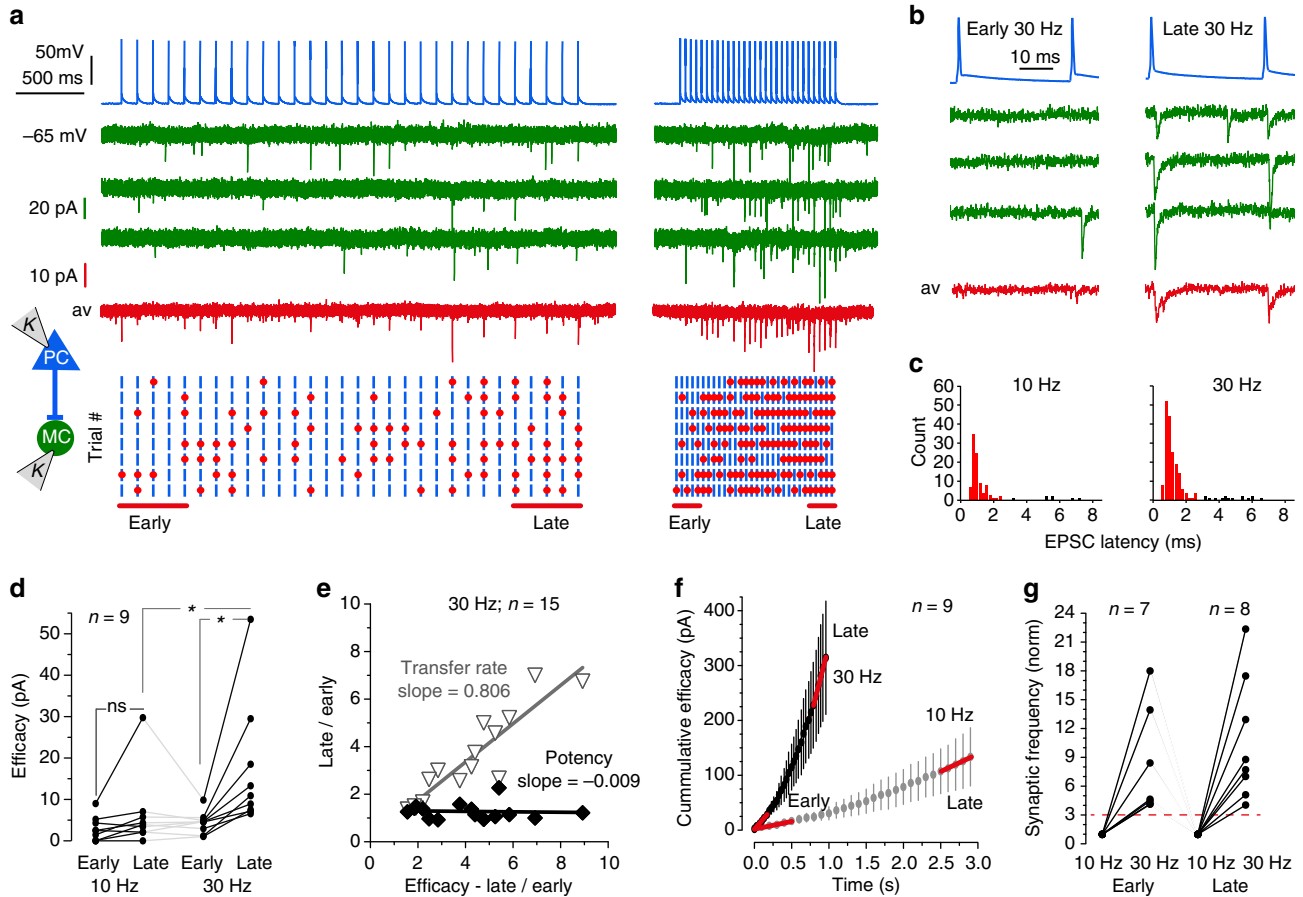

**Figure 5 | Repetitive stimulation progressively unmutes PC–MC excitatory synapses.** (**a**) Thirty action potentials elicited at 10 Hz (left) and at 30 Hz (right) in a PC (blue traces). Voltage clamp traces from a connected MC at − 65 mV. Three successive trials in green, average of eight trials in red. The inter-trial-interval here was 40 s. Below, raster plots of synaptic transfer for eight trains show presynaptic action potentials (blue bars), and EPSCs (red dots) triggered at monosynaptic latencies (0–3 ms). More EPSCs were elicited during the 30 Hz than the 10 Hz train and by late (last five) compared to early spikes (first five). (**b**) Detail of early and late PC spikes and MC-responses in 30 Hz trains. The transmission transfer rate was higher for late stimuli. (**c**) Poststimulus-histogram of EPSCs at monosynaptic latencies, of 0–3 ms, show peaks at 1.16 and 1.14 ms (median) for trains at 10 and 30 Hz. Total counts were higher for 30 than 10 Hz, due to the frequency dependence of release. (**d**) PC–MC synaptic efficacy (transfer rate × absolute potency) showed a strong dependence on spike number during a train and spike frequency ($n = 9$, Friedman test, $P = 0.0002$). Facilitation occurred at 30 Hz but not at 10 Hz (*Dunn's multiple comparison test, $P < 0.05$). (**e**) Late/early transfer rate and potency plotted against late/early efficacy ($n = 15$ pairs, 30 Hz stimulation). Increased efficacy resulted from a higher transfer rate rather than changes in potency. (**f**) Non-linear cumulative efficacy (mean ± s.e.m., summed efficacy over time) plotted against time shows facilitation dynamics and frequency dependence. (**g**) The synaptic frequency increased more than the 3-fold change in presynaptic spike frequency (red dashed line), for both early and late spikes, during 10 and 30 Hz trains. Stimulus artifacts blanked in (**a**,**b**). For detailed statistics, see Supplementary Table 3.

high-frequency firing (from $0.9 \pm 0.6$ to $21.7 \pm 11.9$ pA; $n = 5$). Synaptic unmuting persisted during later sparse firing, even after a silent period of several hundred milliseconds (Fig. 7e,f). We noted both an increase in synaptic events 'locked' to presynaptic spikes with latencies <3 ms, and an increase in delayed excitatory events after sustained high-frequency firing (Fig. 7e). High-frequency pyramidal cell firing could induce late Martinotti-cell discharge consistent with facilitating synapses. Synchronous and asynchronous EPSPs summed to reach Martinotti-cell firing threshold during repetitive high-frequency firing (Fig. 7g,h).

For comparison, we examined the synaptic transmission of the same spike train onto fast-spiking parvalbumin (PV) expressing interneurons in paired PC–PV recordings (Supplementary Fig. 4). PV neurons responded with highest efficacy at the onset of a high-frequency spike train, then displayed depression. Thus a strongly facilitating synaptic transmission is specific to connections made with MC, not PV interneurons.

**Spike timing-dependent inhibitory effect**. We next asked how Martinotti cell-mediated feedback IPSPs affect postsynaptic PC. Pyramidal cell spikes typically initiated Martinotti-cell firing at a latency shorter than 8 ms (84% of spikes, $n = 4$ pairs; Fig. 8a, right panel). Therefore, for reciprocally connected cells most self-induced Martinotti cell-mediated IPSPs occurred with short latency (<10 ms) after a PC spike. These feedback IPSPs coincided with the spike after hyperpolarization (AHP) of the triggering pyramidal cell. Fig. 8a, shows that summation of feedback IPSPs with the AHP resulted in a larger pyramidal cell hyperpolarization enhancing the peak amplitude of the next pyramidal cell action potential, but with little inhibitory effect on pyramidal cell firing.

MC also mediate a lateral inhibition of PC with no reciprocal connection. The timing of IPSPs in these PC is unrelated to previous pyramidal cell firing. Functional differences in the effects of feedback and lateral inhibition on PC firing can be addressed by comparing short (<10 ms) and long latency IPSPs.

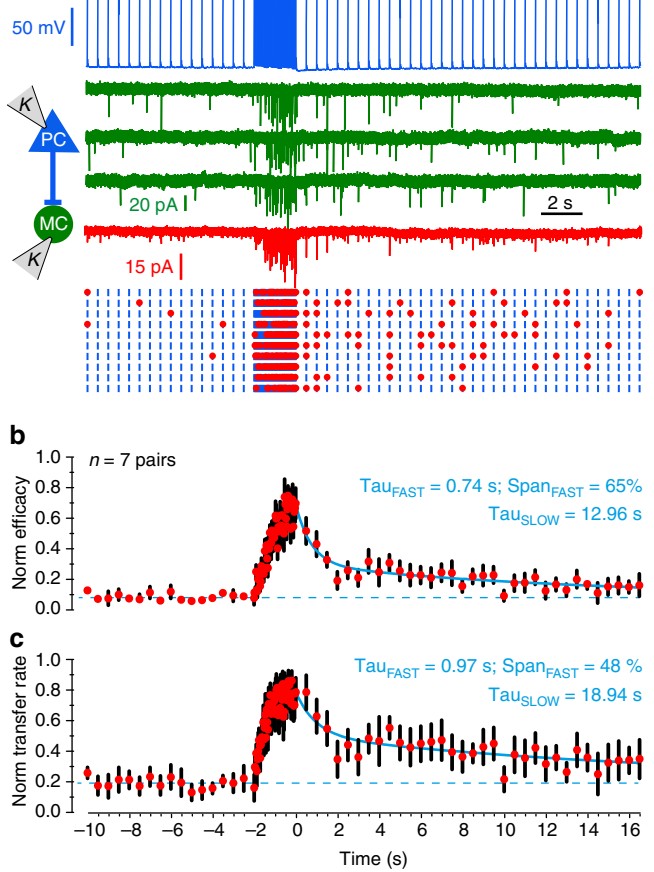

**Figure 6 | Enhancement of PC-to-MC synaptic transmission outlasts train of high-frequency stimulation.** (**a**) Synaptic efficacy and transfer rate were monitored (at 2 Hz) before and after unmuting induced by a 30 Hz presynaptic spike train of 2 s. (**b**) Synaptic efficacy and (**c**) transfer rate for $n = 7$ pairs normalized to the maximal level reached during unmuting and averaged (mean ± s.e.m.). Decay of these parameters was fitted with a dual exponential function (blue line): $\text{Decay} = \text{UM} - \text{Span}_{\text{FAST}} \cdot e^{-\frac{t}{\text{Tau}_{\text{FAST}}}} + \text{Span}_{\text{SLOW}} \cdot e^{-\frac{t}{\text{Tau}_{\text{SLOW}}}}$. $\text{Span}_{\text{FAST}}$ and $\text{Span}_{\text{FAST}}$ indicate the relative contributions of $\text{Tau}_{\text{FAST}}$ and $\text{Tau}_{\text{SLOW}}$. Baseline was constrained to the mean value for the spikes preceding unmuting (broken line).

We tested differences between lateral and reciprocal inhibition in recordings from unidirectionally connected MC-to-PC pairs with mean IPSP amplitude greater than $-0.3$ mV at $-50$ mV ($n = 7$; Fig. 8b–e). MC firing was timed to initiate IPSPs at different times during the PC firing cycle ($n = 7$, 30–50 Hz). We then compared the effects of IPSPs initiated at delays of $<10$ ms (feedback-like) or $>10$ ms after PC firing. We measured peak values for the pyramidal cell AHP ($\text{AHP}_{\text{TEST}}$), the peak of the next spike ($\text{PK}_{\text{TEST}}$) and also interspike interval ($\text{ISI}_{\text{TEST}}$). Since adaptation may occur during spike trains, observed values were compared with the extrapolated values. We found pyramidal cell AHPs were larger for short latency IPSPs ($-0.49 \pm 0.13$ mV, Fig. 8c, Wilcoxon-signed rank test, $P = 0.0313$) than for long latency IPSPs ($0.02 \pm 0.02$ mV; Fig. 8c, Wilcoxon-signed rank test, $P = 0.6875$). Action potential amplitude increased at short latencies ($0.31 \pm 0.11$ mV) compared with delayed MC firing ($-0.06 \pm 0.05$ mV; Fig. 8d). Interspike interval was more prolonged for IPSPs induced at delays $>10$ ms than for short latencies ($111.6 \pm 2.6\%$ versus $103.3 \pm 0.95\%$; Fig. 8e, unilateral

Wilcoxon matched-pairs signed rank test, $P < 0.01$). Thus, the effect of Martinotti-cell IPSPs depends on when they are initiated during a PC firing cycle. Short latency feedback IPSPs, induced when persistent PC firing recruits a MC, may tend to encourage PC firing. In contrast, IPSPs impinging on non-reciprocally connected PCs later in their firing cycle mediate an inhibitory delay of PC firing.

**Inhibitory attractor network model of HD signalling.** Head direction signals are organized, such that neurons with similar preferred head directions fire together in a correlated way[29]. Computational models suggest that this activity profile may emerge from an attractor network[20,23,32,33]. These models mostly rely on strong excitation between cells with similar preferred directions. We asked whether experimentally measured connectivity, strength and dynamic behaviour of synapses in recurrent Martinotti-cell circuits could generate an alternative form of attractor network with no recurrent excitation between PC.

Presubicular pyramidal cell and Martinotti-cell interactions were simulated in a firing rate model, with interneurons and principal cells represented as a two-layer network (Fig. 9a). PC influenced each other exclusively by feedback inhibition, mediated via MC. Each pyramidal cell was assigned a preferred firing direction, so that the entire population evenly spanned 360°. The network was modelled on experimental findings of this study. (1) Each pyramidal unit contacted multiple Martinotti units, and vice-versa, with many reciprocal connections (local dependence on phase similarity of head direction cells; cf. Methods). (2) Excitation of Martinotti units facilitated with a slow decay. (3) Martinotti-mediated inhibition was stable. (4) Martinotti units were spontaneously active. (5) The inhibitory strength of IPSPs was a function of the excitation received by presynaptic pyramidal units. Highly active pyramidal units were little affected by reciprocal feedback inhibition. Stronger, lateral inhibition was exerted on less active pyramidal units (Fig. 9a, right panel). (6) Head direction information transmitted from the thalamus directly activated pyramidal units, but not Martinotti units.

In the absence of correlated inputs, the model network spontaneously generated a directionally selective increase in activity, thus satisfying attractor network dynamics (Fig. 9b). Model neurons coding for a certain direction mirrored directional thalamic input, and when the external drive was reduced and the system relaxed, the neuronal activity profiles were mostly maintained (Fig. 9c). Polar firing plots for representative PC were similar to those of finely tuned head direction cells *in vivo*, while Martinotti-cell firing was little modulated by direction (Supplementary Fig. 5). The precision of the pyramidal cell tuning could be controlled by varying the range $\alpha$ of the inhibition suppression around an existing connection between a pyramidal cell and a Martinotti cell. The model also provided insight on how facilitating synaptic dynamics of Martinotti-cell recruitment contribute to a coherent activity bump. Replacing facilitating synapses with depressing or stable synapses did not suppress bump formation if synapses rapidly returned to their initial state (large b1, Supplementary Fig. 5). However, the correlation of principal cell firing with initial external inputs was considerably degraded. Facilitating PC-to-MC connections seem then to be crucial to maintain head directional signals (Fig. 9d,e). Recurrent excitation of PV interneurons, which exhibits a dynamic depression, may not be needed to maintain head directional information in the presubiculum. In conclusion, inhibitory feedback activated at high firing frequencies, and with effects dependent on IPSP timing during the pyramidal cell

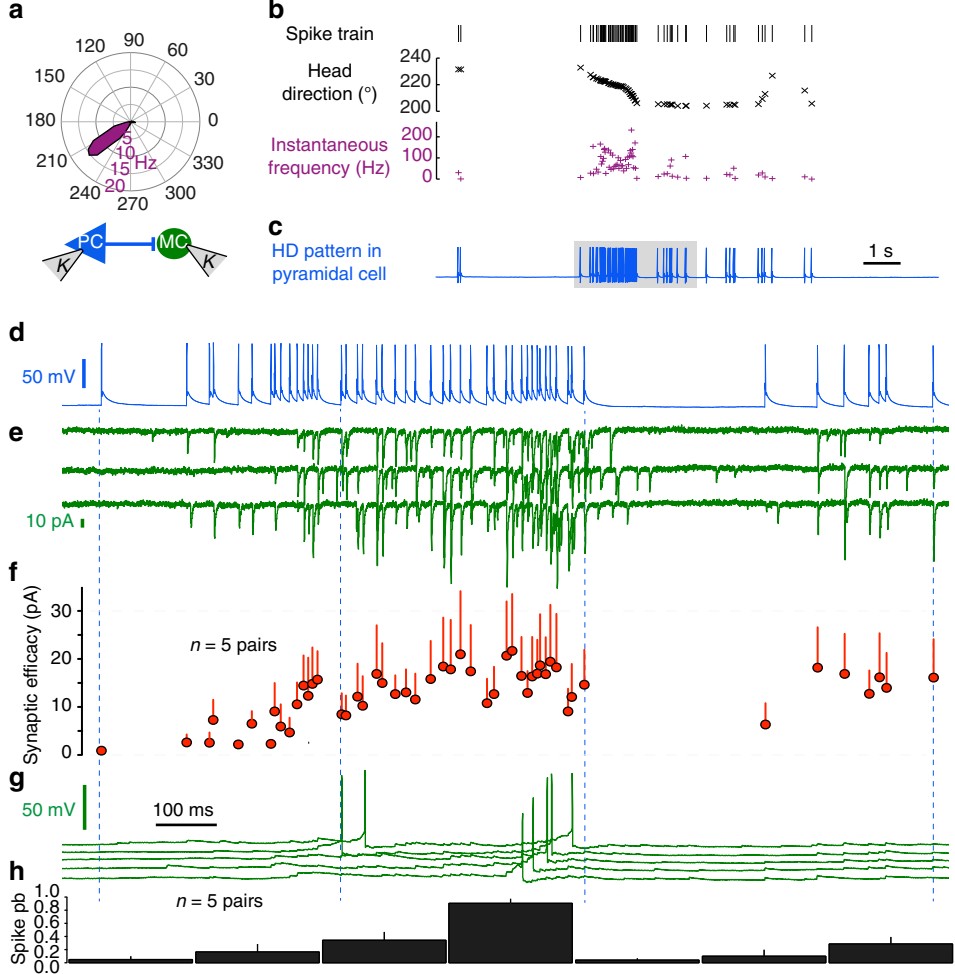

**Figure 7 | PC-to-MC synaptic unmuting and Martinotti-cell recruitment by head direction spike trains.** (**a**) Polar plot showing firing frequency (spike/sec) as a function of head direction (°) for a head direction cell recorded *in vivo*. (**b**) Detected spikes for this unit plotted against instantaneous head direction and frequency. (**c**) The spike train was injected as a current command into a presubicular pyramidal cell *in vitro*. (**d**) The shaded part of the trace in **c** extended (total trace time 1.4 s). (**e**) Three successive responses recorded from a connected MC in voltage clamp at −65 mV (green traces) show unmuting and facilitation. (**f**) Synaptic strength for EPSCs from five pairs. Transfer was null at first and increased progressively after sustained high-frequency firing. Transfer rate remained high during subsequent lower frequency firing (at right). (**g**) Current clamp recording of MC cell in **e** at resting potential. Four successive traces show reliable firing towards the end of high-frequency head direction derived spiking. (**h**) MC firing probability (200 ms bins) was maximal after persistent high-frequency PC firing (from 5 pairs). Error bars represent sem.

firing cycle, can suffice to refine and sustain presubicular head direction signals.

## Discussion

We have described activity-dependent dynamic properties of the Martinotti-cell inhibitory feedback loop in the presubiculum. These properties underlie a self-sustained processing of head direction information in presubicular microcircuits. Superficial PC are directly excited by thalamic inputs. Martinotti type interneurons are excited by these PC and reliably inhibit pyramidal cell dendrites in layers 1 and 3. Feedback excitatory transmission from PC-to-MC is greatly facilitated during sustained high-frequency presynaptic firing. Synaptic transfer may be enhanced for several seconds after a PC-to-MC connection is 'unmuted'. The behaviour of this feedback inhibitory circuit is directly relevant to patterns of head direction activity. Natural firing patterns of these cells, recorded *in vivo*, recruited MC very effectively *in vitro*, whereas lower firing frequencies had little effect. Firing of these interneurons had

distinct timing-dependent effects. In reciprocal connections, MCs fired at short latencies after PC action potentials. Inhibition by such precisely timed, spike-locked IPSPs was less effective than for randomly timed IPSPs, such that MC provide a strong lateral inhibition. This feedback circuit is well-adapted to refine head direction signals in the presubiculum and to robustly preserve sustained firing of in-tune head direction cells.

Head direction signals are thought to be generated in subcortical nuclei and relayed via the thalamus to the parahippocampal region[1,6]. Neurons of anterior thalamus (ATN) project quite specifically to the presubiculum[7] (Fig. 2). A monosynaptic connection from ATN to presubicular head direction cells has been recently inferred *in vivo* based on short latency, reliable spike transmission[29]. Here, we examined the effects of optogenetic activation of anterior thalamic axon terminals on single presubicular neurons *in vitro*. Our data provide functional evidence for a direct innervation of layer 3 pyramidal neurons of the presubiculum by thalamic fibres. Martinotti type interneurons received no direct excitation. PC of superficial layers project directly to the MEC[10,11,34,35]. While grid-cell activity of MEC

neurons depends on head direction information[36], the ATN does not project directly to the MEC. Thus, integration of head direction code in presubicular superficial layers seems to be an essential element in the construction of inputs to MEC grid cells.

Recurrent feedback circuits of MC and PC are highly interconnected. The probability of PC-to-MC connections was 37%. The MC-to-PC connection probability was even higher: 58%. Such estimates from paired recordings are probably underestimates since all connections may not be preserved in

slices. Our pipette solution was designed to artificially enhance the driving force for chloride, by using low chloride in the internal solution, increasing our ability to detect inhibitory synaptic events and to distinguish them from failures. Nevertheless, we may have missed low amplitude inhibitory synaptic events generated at very distal dendritic sites. MC of other cortical areas also have high connection probabilities with local PC to provide a dense, reliable and non-specific inhibition[37], with both convergent and divergent connectivity[25,38]. We detected no direct activation of MC by thalamic afferents reinforcing the feedback role of MCs in a presubiculum circuit. With a very low rate of recurrent connection between PC ($\sim$2%), the PC–MC pathway becomes especially important to mediate interactions between presubicular PC, similar to layer 5 PC in neocortex[25] or to layer 2 stellate cells in medial entorhinal cortex[39,40].

We found MCs were only excited to fire by summed EPSPs induced after synaptic unmuting when PCs fired at high frequencies for prolonged periods. Single PC spikes never led to MC discharge (Figs 7g and 8a). The short-term dynamics of pyramidal cell synapses vary between fast-spiking, parvalbumin expressing or low threshold spiking, somatostatin expressing interneurons in neocortex and hippocampus[25,41–43] (cf. also Supplementary Fig. 4). The facilitation during repeated activation shown here at synapses that excite MC is similar to that of synapses made with SST immunopositive interneurons in hippocampus[42] and neocortex[24,25,43,44], even though the presubiculum is not a typical neocortical area, but rather part of the transitional periarchicortex. Synaptic facilitation in the presubiculum has slow kinetics, corresponding well to the persistent discharges of head direction cells. In somatosensory cortex, a threefold increase in average EPSP amplitude could be obtained after eight stimulations at 20 Hz (ref. 44), while in presubiculum a similar degree of facilitation was obtained after 30 stimuli at 30 Hz. Presubicular PC-to-MC synapses were often silent during paired pulse stimuli.

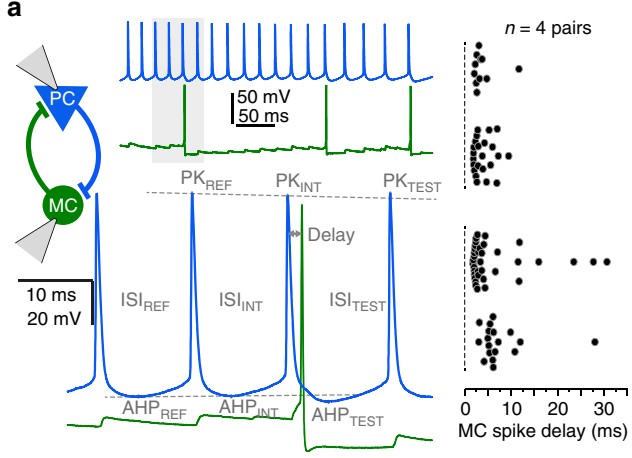

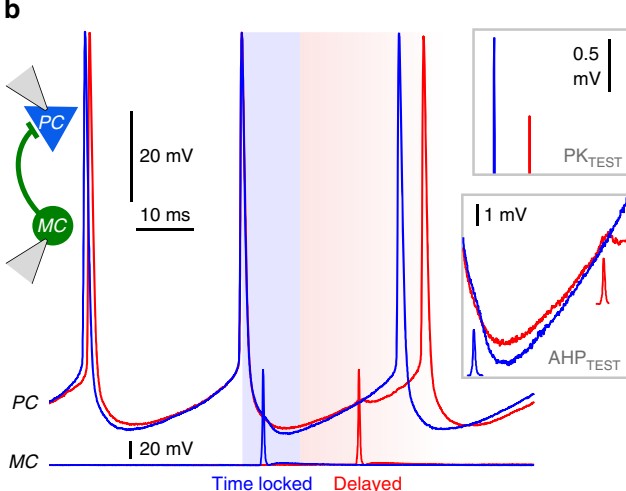

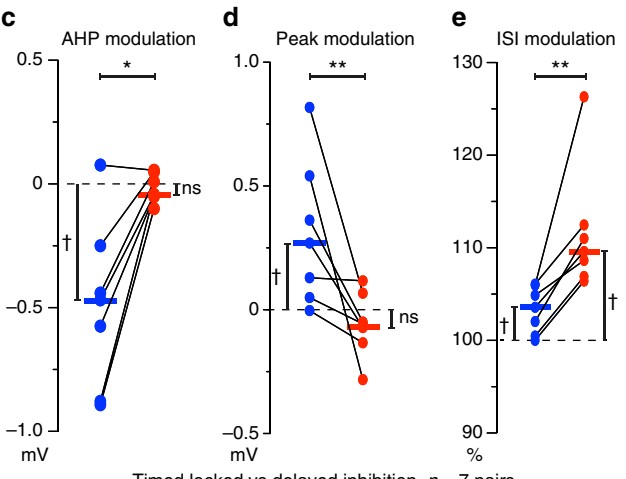

**Figure 8 | Timing dependence of inhibition. (a)** MC recruitment by a PC in a reciprocally connected pair (top left). Magnification of a MC-spike evoked by PC firing at short latency (bottom). The MC-spike alters the PC-spike AHP (AHP$_{TEST}$), the PC-spike peak (PK$_{TEST}$) and the PC ISI (ISI$_{TEST}$) according to the delay after the PC-spike PK$_{INT}$. Dotted lines indicate the extrapolated level for PK$_{TEST}$ and for AHP$_{TEST}$. Right, MC firing triggered by PC spikes in $n = 4$ PC–MC pairs. 84% of MC spikes have a delay of <8 ms after a PC spike (65 out of 77 spikes; $n = 4$ pairs). **(b)** Spike timing-dependent MC inhibition was tested in unidirectionally connected MC-to-PC pairs. Drifting single MC spikes were triggered during sustained PC firing (30–50 Hz). Two sweeps of PC firing are shown (one in blue, one in red), with the corresponding MC spikes at the bottom. For the blue voltage trace, the PC-spike to MC-spike delay was short ('time locked'; delay <10 ms, similar to reciprocal connections as in **a**). For the red voltage trace ('delayed'), the MC-spike delay exceeded 10 ms. **(c–e)** Differential effect of short latency versus long latency inhibitory modulation of AHP, peak and ISI of PC spikes (30–50 Hz; $n = 7$). **(c)** The PC AHP was more hyperpolarized for short latency, time-locked MC spikes (blue) but not for delayed (red) MC spikes ($n = 7$, *$P < 0.05$). The modulation of the PC-spike AHP was calculated as (AHP$_{TEST}$ − AHP$_{INT}$) − (AHP$_{INT}$ − AHP$_{REF}$). **(d)** The PC-spike peak after a MC-spike was higher for timed locked but not for delayed inhibition ($n = 7$, **$P < 0.01$). Peak modulation was calculated as (PK$_{TEST}$ − PK$_{INT}$) − (PK$_{INT}$-PK$_{REF}$). **(e)** The PC ISI increased more for delayed than for time-locked inhibition ($n = 7$, **$P < 0.01$). ISI change was calculated as $100 \times$ (ISI$_{TEST}$/ ISI$_{INT}$)/(ISI$_{INT}$/ISI$_{REF}$). Each dot indicates the mean for one pair. Horizontal bars are medians. The median-null difference was assessed with a Wilcoxon-signed rank test ($^{\dagger}P < 0.05$) and the relative difference between short- and long-latency inhibition with a Wilcoxon matched-pairs signed rank test (*$P < 0.05$, **$P < 0.01$). AHP, after hyperpolarization; ISI, interspike interval; PK, peak.

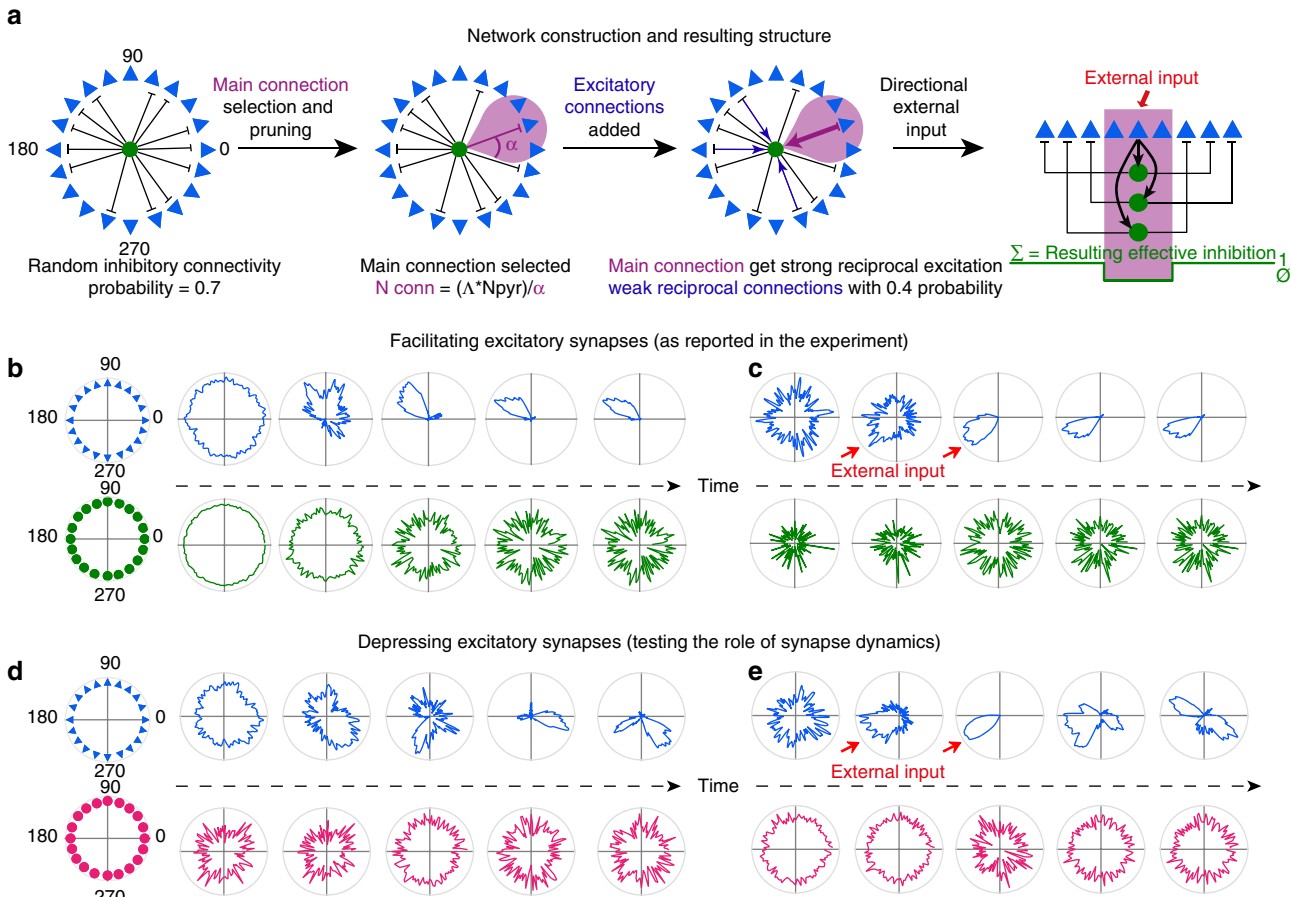

**Figure 9 | Structure and activity patterns of the neural network model.** (**a**) Schematic representation of the network wiring process. Left: Initial random connectivity between each Martinotti cell (green circle) and the pyramidal cell population (blue triangles). Middle panels: Establishment of final connectivity by selecting and strengthening the main connections (for simplicity only 1 main connection is shown; inhibitory strength $\lambda = 0$). Connections with neighbouring units are removed (pink area), reciprocal excitatory connections are added (blue). Right: resulting inhibitory effect of each pyramidal cell on the rest of the population. Phase selective suppression of activity is mediated by collective Martinotti-cell activity. (**b–e**) The dynamic properties at the excitatory PC-to-MC synapses are governed by two parameters, $b1$, controlling the persistence of synaptic modulation, and $b2$, controlling the strength and direction of the modulation (cf. Methods and Supplementary Fig. 5). To examine the importance of facilitating synapse dynamics, $b1$ and $b2$ are chosen to model either facilitating ((**b,c**) $b1 = 0.00025$; $b2 = 0.5$) or depressing excitatory synapse dynamics ((**d,e**) $b1 = 0.04$; $b2 = -0.5$). (**b**) Spontaneous activity profile formation in PC. In blue, example of raw activity over time of the pyramidal cell population (arranged according to directional selectivity) in the absence of selective external input. Fed with white noise input, the system evolves over time from an initial random configuration (left) towards a stable configuration of concentrated activity ('bump' formation, right). In green, same as above, but for simultaneous activity of MC (positions on the circle are randomly assigned). (**c**) Persistence of directional tuning after external input removal. Starting from random activity (left), PC around a given direction are transiently provided with an additional external input (red arrows). The final state of the network is then observed after removing this additional input (grey background). No directional selectivity appears in MC. (**d**) Replacing facilitating by depressing dynamics at excitatory synapses impairs the spontaneous development of a coherent activity bump in the network. (**e**) Depressing excitatory synapse dynamics do not support the persistence of directional activity correlated to an initial external input.

We therefore analysed synaptic dynamics from responses to trains of action potentials at 10 or 30 Hz. Enhanced synaptic efficacy during these trains resulted from increased transfer rate rather than potency (Fig. 5e). This phenomenon persisted for a time after high-frequency stimuli (Fig. 6; Supplementary Fig. 3) as at some other synapses[31,45]. Nevertheless even after unmuting, the transfer rate remained quite low at this synapse, compared to responses elicited by similar stimuli at neocortical PC-to-MC synapses in layer 3 (ref. 44) or layer 5 (ref. 25).

Possibly presubicular PC-to-MC transmission is regulated by an activity-dependent mechanism, distinct from short-term facilitation[31,46], situated at either axonal or presynaptic sites[47], and affecting spike waveform or the release machinery[26,48,49]. Transfer rate depends on both the probability of neurotransmitter

release $p_r$, and the number of release sites. Data on numbers of terminals and active zones as well as the location of synapses on postsynaptic membrane will necessitate ultrastructural work. Further, since basal and dynamic values for $p_r$ may differ at individual PC–MC synaptic boutons[50], a full description would also require information on vesicle pool size and replenishment. This work has rather presented an average picture of PC–MC synapses. Functionally, activity-dependent synaptic unmuting and asynchronous release provide a medium-term synaptic memory[51]. Such a trace of recent head directions at this synapse would permit comparison with incoming visual and hippocampal information converging in the presubiculum.

About half of presubicular principal neurons are directionally modulated[3]. Head direction cells sustain firing at high frequencies

with weak adaptation while an animal maintains its head in a preferred direction[4]. PC-to-MC synapses are perfectly tuned to activate recurrent inhibition for sustained HD signals. With very low initial transfer rates they act as a high pass filter[30], insensitive to sharp increases, but enhanced over time. Unlike fast-spiking parvalbumin expressing interneurons[10], MCs may therefore not be recruited during fast head turns when head direction cells do not fire persistently. MC inhibition of pyramidal cell dendrites seems likely to control the genesis of regenerative dendritic events[27]. It will act to suppress dendritic electrogenesis and counter the effects of EPSPs impinging at local dendritic sites. MC feedback inhibition would prevent over-excitation and control network activity[19,52] during persistent head direction firing. But it would also provide a common time window for dendritic excitation and so synchronize firing in neurons with similar head direction tuning. In the context of head direction sensitivity, spiking output is a critical variable, and head direction signalling is identified *in vivo* according to neuronal firing. The influence of MC inhibition on somatic and axonal processes of spike generation may be relatively minor. Nevertheless, MCs may affect somatic firing by propagation of a dendritic hyperpolarization or by preventing the somatic propagation of dendritic EPSPs. Further work is needed to experimentally test how, in presubiculum, inhibitory and excitatory inputs impinging on postsynaptic PC dendrites are integrated. The effects of inhibition vary with timing during the pyramidal cell firing cycle (Fig. 8; refs 53,54). In reciprocally connected cell pairs, when PCs drive MC firing, an IPSP is generated at short latency during the PC AHP. These IPSPs enhance repolarization which may facilitate initiation of the next PC action potential. Short latency feedback inhibition is therefore functionally less inhibitory than randomly timed lateral inhibition. Because well-tuned head direction cells fire maximally, and poorly tuned cells fire less[14], PC-to-MC synapse dynamics clearly favour MC recruitment by well-tuned direction cells. Lateral feedback then preferentially inhibits poorly tuned cells.

Experimental data let us propose a modified continuous attractor model based on recurrent inhibition to mimic head direction activity. The build-up of strong principal neuron activation as a necessary condition for interneuron recruitment is essential to the model. We suggest that activity-dependent unmuting of MC and their facilitating synapse dynamics are key for autonomous circuit dynamics in the presubiculum. In contrast to primary visual cortex[55], the presubiculum could, in this way, sustain activity. Fast-spiking PV neurons with depressing synapses (Supplementary Fig. 4) are not part of the attractor. We suggest that during fast head turns, when MC are not recruited, the system may switch to a relay type function. Presubicular fast-spiking interneurons fire at higher rates during rotation[10], that is, when the population of active head direction cells shifts quickly. The excitatory inputs received by PV neurons continuously changes to different sets of synapses, and for each transient head direction, PV neurons will rapidly provide inhibition with depressing dynamics. However, MC become active and stay active during maintained directional signalling, and could support a form of working memory[56–58], especially in the absence of a stabilizing stimulus[55]. Our data on a one-dimensional head direction system might suggest that equivalent dynamics exist in the medial entorhinal grid-cell system[39,59]. We note the model network requires no directional tuning of presubicular interneurons. The efficacy of inhibitory synapses depends exclusively on the timing of interneuron firing with respect to firing in the presynaptic principal neuron, leading to little inhibition for a driving head direction cell, but stronger lateral inhibition. In conclusion, the recruitment of MC by differentially active, randomly connected PC provides an economic way to refine and sustain presubicular head direction signal representations.

## Methods

**Animals.** Most work was done on slices from 4 to 7 weeks old male and female transgenic mice (X98-SST line, JAX 006340) that express GFP in a subpopulation of somatostatin-positive (SST) neocortical MC[28]. X98-SST mice were maintained by breeding heterozygous males with C57BL/6J females (CERJ Janvier). Sst-Cre::tdTomato mice were used in experiments involving the light activation of Channelrhodopsin-2 (ChR2). They were obtained by crossing male Sst-IRES-Cre mice[60] (Jax 013044) with females from the Ai14 Cre reporter line[61] (Jax 007914). SST positive neurons of these mice express the red fluorescent protein tdTomato, which can be visualized without activating ChR2. Pvalb-Cre::tdTomato mice were used for comparing synapse dynamics. They were obtained by crossing Pvalb-Cre mice[62] (Jax 008069) with the Ai14 Cre reporter line. Animal care and use conformed to the European Communities Council Directive of 2010 (2010/63/EU) and French law (87/848). The animal protocol (no. 01025.02) was approved by the local ethics committee Charles Darwin No. 5 and the French Ministry for Research.

**Stereotactic virus injections.** Adeno-associated viral vectors carrying genes for ChR2-EYFP fusion proteins (AAV2/9.hSyn.hChR2 (H134R)-EYFP.WPRE.hGH; University of Pennsylvania Vector Core) were injected into the ATN at postnatal age P28. For surgery, mice were deeply anaesthetized with intraperitoneal injection of ketamine hydrochloride and xylazine (100 and 15 mg kg$^{-1}$, respectively) following stereotaxic procedures described previously[63]. The virus was delivered via a 33-gauge needle with a Hamilton syringe in a syringe Pump Controller (Harvard Apparatus, Pump 11 elite) at 20 nl min$^{-1}$. ATN was targeted at coordinates from Bregma: lateral 0.75 mm; posterior, $-0.82$ mm; depth, $-3.2$ mm. Slices were prepared at 12–16 days after vector injection. The injected volume was 150 nl, to be as specific as possible (cf. Supplementary Fig. 1), but with enough spread to cover ATN.

**In vitro electrophysiology and photostimulation.** Under ketamine and xylazine anaesthesia, animals were perfused via the heart with 30 ml or more of a solution containing (in mM): 125 NaCl, 25 sucrose, 2.5 KCl, 25 NaHCO$_3$, 1.25 NaH$_2$PO$_4$, 2.5 D-glucose, 0.1 CaCl$_2$ and 7 MgCl$_2$, cooled to 2–6 °C and equilibrated with 5% CO$_2$ in O$_2$. The forebrain was dissected, and horizontal slices of thickness 260–320 μm were cut with a vibratome (Leica VT1200S). They were transferred to a storage chamber containing warmed (37 °C) artificial cerebrospinal fluid (ACSF) of: 124 NaCl, 2.5 KCl, 26 NaHCO$_3$, 1 NaH$_2$PO$_4$, 2 CaCl$_2$, 2 MgCl$_2$ and 11 D-glucose (mM), gently bubbled with 5% CO$_2$ in O$_2$ (pH 7.3, 305–310 mOsm l$^{-1}$). ACSF in the storage chamber cooled towards room temperature (22–25 °C) as slices were kept for at least 1 h before transfer to a recording chamber.

The recording chamber, of volume ~2 ml, was heated to 33–35 °C. Neurons were visualized with an EMCCD Luca-S camera (Andor) on an Axioskop 2 FS plus microscope (Zeiss, France) with infrared differential interference contrast. Glass recording pipettes were pulled from borosilicate glass of external diameter 1.5 mm (Clark Capillary Glass, Harvard Apparatus) using a Brown-Flaming electrode puller (Sutter Instruments). A low-chloride potassium gluconate-based (Low-Cl K-gluc) internal solution contained (in mM): 145 K-gluconate, 2 KCl, 10 HEPES, 0–0.2 ethylene glycol tetra-acetic acid (EGTA), 2 MgCl$_2$, 4 MgATP, 0.4 Tris-GTP, 10 Na$_2$-phosphocreatine. The caesium gluconate-based internal solution (Cs-gluc) contained (in mM): 135 Cs-gluconate, 5 KCl, 10 HEPES, 0–0.2 EGTA, 2 MgCl$_2$, 4 MgATP, 0.4 Tris-GTP, 10 Na$_2$-phosphocreatine. Recordings were made with low-Cl K-gluc solution unless specified. Tip resistance of filled pipettes was 3–7 MΩ. Whole-cell records were made with a Multiclamp 700B amplifier and acquired with pClamp software (Molecular Devices). Recordings were filtered at 6–12 KHz in current clamp mode and at 2–6 KHz in voltage clamp mode. No correction was made for junction potential (~15 mV). Access resistance was continuously monitored and records were excluded if variations exceeded 15%.

PC were identified as non-fluorescent regular spiking neurons with typical properties[15]; Martinotti-like cells (MC) of tissue from X98-SST mice were defined as green fluorescent neurons, and those from Sst-Cre::tdTomato mice as red fluorescent neurons. In both mouse lines, MC possessed resting membrane potential above $-65$ mV. Discharges were either adapting or low threshold firing and biocytin filling revealed typical Martinotti-cell axonal and dendritic morphologies (Fig. 2).

Channel rhodopsin-expressing terminals from the AT thalamic nucleus were excited with blue light from a source (Cairn OptoLED, white) coupled to the epifluorescence microscope port, filtered (BP 450–490, FT 510) and fed into a ×60 1.0 NA plan-Apochromat objective. Light pulses of 0.5 ms duration and intensity 2 mW were delivered at 20 s intervals. Overall, 1 μM TTX and 40 μM 4AP were added to the bath to check for direct versus indirect optical activation. Salts and anaesthetics were all obtained from Sigma, except TTX from Tocris.

**In vivo electrophysiology.** Head direction firing was sampled from presubicular neurons *in vivo* in rats, with higher channel counts and unit yield compared to mice (cf. ref. 3). Even though inter-species differences in network microstructure may exist, HD direction cells have been described in presubiculum in mice and in rats[3,4,10,29], and intrinsic neuronal properties and projection targets are similar in both species[15,34,64]. Briefly, tetrodes were implanted in 4 months old Long-Evans rats at AP 2.2 mm in front of the transverse sinus, ML 3.7 mm from the midline,

and DV 1.5 mm below the dura. Tetrodes were lowered progressively until reaching presubicular layers. Recording sites in presubiculum were confirmed from *post hoc* Nissl, parvalbumin and calbindin stained sections. Head direction was tracked with two light-emitting diodes while the animal collected randomly distributed food crumbs from a 100 cm wide square box. Spikes were sorted offline with cluster cutting Axona software. Head direction was calculated from projections of the relative position of the two LEDs on the horizontal plane. Directional tuning for each cell was obtained by plotting firing rate against the rat head direction, divided into bins of 3° and smoothed with a 14.5° mean window filter (14 bins on each side). Command protocols for slice records were generated from these spike trains imported into pClamp.

**Data analysis.** Signals were analysed with AxoGraphX, and locally-written software (Labview, National Instruments; MATLAB, The Mathwork). Algorithms to detect action potentials and measure active and passive neuronal properties were described previously[15,16]. Given are mean ± s.e.m. unless otherwise stated.

**Detection of postsynaptic events.** Excitatory and inhibitory postsynaptic currents and potentials were detected and measured automatically from low-pass filtered records adapted to the recording mode (0.4 KHz for EPSPs, 1 KHz for EPSCs and 500–750 KHz for IPSCs). Spontaneous or spike-associated events were detected as continuous rising signals exceeding a threshold set for records from each cell to minimize both false positive and negative detection. Thresholds were 0.3–0.6 mV for EPSPs, 4–7 pA for EPSCs and IPCSs recorded in K-gluconate and 4–12 pA for IPSCs recorded with Cs-gluconate solution.

Spike-locked postsynaptic events were defined as first events occurring within a monosynaptic latency (generally 0.5–3 ms for an EPSC and 0.5–4 ms for an IPSC). Delayed postsynaptic events were those that occurred later than the spike-locked events or outside the monosynaptic window but still within 10 ms after the spike. PSC latencies were calculated from the action potential peak to the mid-rise of the postsynaptic event.

Spontaneous activity can bias values for synaptic transfer. We estimated 'false positives' which might exaggerate monosynaptic transfer rates. Presynaptic firing patterns were aligned to a 'control window', before stimulation, and transfer rate, corresponding to a noise value, was calculated. This procedure was applied multiple (250–300) times using different starting points in the same control window. The number of 'false positive' rarely exceeded 0.05. It depended on the level of background synaptic activity, but not on presynaptic firing frequency.

**Synapse dynamics in repetitive stimuli.** Synaptic transfer rate was calculated from paired records as the number of detected postsynaptic events divided by the number of presynaptic spikes. Failure rate was 1—transfer rate. Synaptic potency (pA or mV) was defined as the amplitude of detected events. Efficacy (pA or mV) was the mean amplitude of responses including failures (failure amplitude = 0). Efficacy may be deduced as potency × transfer rate. Synaptic transmission during repeated presynaptic activation was analysed in these terms to derive transfer rate, potency and efficacy for either (1) a given spike across different trials of a standard stimulus, or (2) groups of successive spikes elicited during a defined time. For spike trains, the first five spikes and the last five spikes were grouped to increase measurement precision (Figs 4 and 5) when trial-to-trial variability was high. Changes of transfer rate, potency and efficacy over time are measured as the ratio of late/early values. Cumulative efficacy was calculated as the sum of efficacy over time and provides a temporal dynamic. The derivative of this cumulative efficacy, that we called 'synaptic frequency', corresponds to the information transferred per second.

**Inhibitory effect.** Functional MC-mediated inhibition was quantified as the ability of an IPSP to delay PC-discharge (ISI modulation). We also measured effects of IPSPs induced after pyramidal cell firing. In an effective recurrent circuit, PC may induce MC-spike firing evoking in turn an IPSP in the initiating pyramidal cell. This effect was quantified as an enhanced pyramidal cell AHP (AHP modulation) or change in peak amplitude of a PC-spike (Peak modulation). Both parameters could be affected by intrinsic properties such as adaptation, peak accommodation and an AHP depolarization during repetitive firing. We therefore determined the effect of inhibition as changes from predicted pyramidal cell-repetitive firing behaviour (Fig. 8).

**Cellular anatomy.** Biocytin (1 mg ml$^{-1}$) was added to the pipette solution to reveal the morphology of some recorded cells as described[15,16]. Axo-dendritic morphology was reconstructed from *z*-stacks of acquired images with Neurolucida software (Microbrightfield, Williston, VT, USA).

**Computational model.** Our model builds on previous network models of the head direction system that generate attractor dynamics with directionality[21,65,66]. In the present model, network function is dominated by indirect, inhibitory interactions between PC, and we explicitly include interneurons mediating such interactions in the dynamics of the system.

We simulated the activity of a layer of $N_{pyr}$ pyramidal units interacting through a population of $N_{inh} = \rho N_{pyr}$ Martinotti units. Each pyramidal unit was assigned a preferred direction $\Theta$, evenly spaced to cover the $0$–$2\pi$ interval, while Martinotti units were identified with an index $j$. Pyramidal units did not interact directly through recurrent excitatory connectivity, but they were highly connected to the Martinotti-cell layer and so influenced each other via disynaptic feedback inhibition. The strength of this feedback inhibition exerted from each pyramidal unit on the rest of the population defined the effective interaction among PC. The wiring of the system was conceived to produce a regularly patterned effective connectivity. As in previous continuous attractor models in head direction systems[20–23] connectivity depended on the angular distance of the preferred direction of pyramidal units. Pyramidal units with similar preferred phase did not inhibit each other, while they tended to suppress the activity of units with different directional selectivity.

Excitatory connections from pyramidal to Martinotti units ($W_{\Theta,j}^{OUT}$) and inhibitory connections from Martinotti units back to the pyramidal layer ($W_{j,\Theta}^{IN}$) were established as follows (cf. Fig. 9a). Initially, each Martinotti unit randomly connected to pyramidal units with a 0.7 probability. A sub-set $N_{Conn}$ of these inhibitory connections was randomly selected as 'main connections' with strength $\tilde{w}^{OUT}$. Inhibitory connections contacting neighbouring PC were pruned. Consequently, for each main connection between a Martinotti unit $j$ and a pyramidal unit $\Theta$, connections from the Martinotti unit to pyramidal units with directional preference close to $\Theta$ were pruned. The number of main connections $N_{Conn}$ for each Martinotti unit depended on the range of pruning, $\alpha$. The number of pruned connections for each main connection was a constant fraction $\Lambda$ of all the possible connections for any range of the pruning, that is $\alpha N_{Conn} = \Lambda N_{pyr}$. Therefore:

$$N_{Conn} = \frac{\Lambda * N_{Pyr}}{\alpha} \tag{1}$$

After establishing the inhibitory connectivity, the excitatory wiring was established as follows. Each main connection was associated with a reciprocal excitatory connection of uniform strength. Additional excitatory connections were created with a 0.4 probability and a reduced strength, compared to those associated with the main connections, randomly drawn from the $[0\ w^{OUT}]$ interval (where, $w^{OUT} < \tilde{w}^{OUT}$).

As a last step, the strength of the inhibitory connections converging on each pyramidal cell was normalized according to:

$$\sum_j W_{j,\Theta}^{IN} = w^{IN} \tag{2}$$

Ultimately, three different groups of connections could be found (cf. Fig. 9a): (1) strong main excitatory connections from pyramidal to Martinotti (purple arrow), (2) weaker excitatory connections (blue arrows) and (3) inhibitory connections (Network parameters are given in Supplementary Table 4).

A pyramidal unit assigned with preferred direction $\Theta$ was described by its firing rate at time $t$, $r_E(\Theta,t)$, regulated through the following dynamics:

$$\tau_E \dot{r}_E(\Theta,t) = -r_E(\Theta,t) + f(h(\Theta,t) + I(\Theta,t)) \tag{3}$$

where $f[I] = g[I]_+$ is a threshold linear $f$–$I$ curve and $\tau_E$ is the neuronal time constant. The input to the unit consisted of an external input term, $h$ (see below), and the contribution coming from feedback inhibition, $I$. In turn, the inhibition term consisted of the combined effect of the presynaptic Martinotti units:

$$I(\Theta,t) = \sum_j w_{j,\Theta}^{IN} r_I(j,t) \tag{4}$$

where $w_{j,\Theta}^{IN}$ is the strength of the inhibitory connection between Martinotti unit $j$ and pyramidal unit $\Theta$. Similarly, for Martinotti units firing rate was regulated by the equation

$$\tau_I \dot{r}_I(j,t) = -r_I(j,t) + f(E(j,t)) \tag{5}$$

that includes a time constant $\tau_I > \tau_E$ generating slower input integration times. For the pyramidal units, the excitatory current $E$ was the sum of presynaptic inputs. For Martinotti units, the summation was modulated by synaptic temporal dynamics described in the next paragraph. Synaptic facilitation may be crucial to stabilize the network[56–58].

The formation of a coherent bump of activity crucially depended on the stable excitation of Martinotti interneurons. For Fig. 9d,e, in order to examine the influence of synapse dynamics on the stability of Martinotti interneurons activity, we modified the previous equation

$$\tau_I \dot{r}_I(j,t) = -r_I(j,t) + f(\tilde{E}(j,t)) \tag{6}$$

so that the term $\tilde{E}(j,t)$ reflected either depressing or facilitating synaptic dynamics. We introduced a variable regulating the responsiveness of synaptic contacts between each pyramidal and Martinotti cell:

$$\tilde{E}(j,t) = \sum_j \gamma_{\Theta,j}^{Eff}(t) w_{\Theta,j}^{OUT} r_E(\Theta,t). \tag{7}$$

The evolution of the synaptic efficacy $\gamma_{\Theta,j}^{\text{Eff}}$ was mediated by the equations:

$$\gamma_{\Theta,j}^{\text{Eff}}(t) = \frac{1}{1 + e^{-k_1(\gamma_{\Theta,j}^{\text{Act}}(t) - k_2)}} \qquad (8)$$

$$\gamma_{\Theta,j}^{\text{Act}}(t) = \gamma_{\Theta,j}^{\text{Act}}(t-1) + b2 * r_E(\Theta, t-1) + b1 * ((1 - \text{sgn}(b2))/2 - \gamma_{\Theta,j}^{\text{Act}}(t-1)) \qquad (9)$$

$\gamma_{\Theta,j}^{\text{Act}}$ is a variable in the range [0,1] that integrates over time the synaptic activity between pyramidal cell $\Theta$ and Martinotti cell $j$. The activation state of the synapses is then turned into the efficacy value $\gamma_{\Theta,j}^{\text{Eff}}$ through a sigmoid transfer function. The two activation parameters control the direction ($b2$) and the persistence ($b1$) of the modulation. If $b2 > 0$ then synapses are facilitating, otherwise, when $b2 < 0$, the equations represent synaptic depression. For increasing $b1$, synapses come back faster to their initial state.

The experimentally observed spike-timing dependence of recurrent inhibition is implemented as the dynamic regulation of the effectiveness of feedback inhibition from a Martinotti unit to a pyramidal unit: Feedback inhibitory strength decreases as the contribution of a given pyramidal cell to Martinotti-cell firing increases. The reciprocal feedback inhibition for each pair was re-computed at every time-step according to:

$$W_{j,\Theta}^{\text{IN}} \rightarrow (1 - R_{\Theta,j}) W_{j,\Theta}^{\text{IN}} \qquad (10)$$

$$R_{\Theta,j} = \frac{w_{\Theta,j}^{\text{OUT}} r_E(\Theta, t)}{\sum\limits_j w_{\Theta,j}^{\text{OUT}} r_E(\Theta, t)} \qquad (11)$$

Therefore, each pyramidal cell would see a modification of the inhibitory connection strength depending on its contribution to the activity of the corresponding Martinotti cell. In the case of a single pyramidal cell driving a Martinotti cell, its feedback inhibition would be zero.

Each pyramidal unit in the network received an independent, time-dependent, activation current $h(\Theta,t)$ from an external source. Since all the internal effective interactions between pyramidal units were inhibitory, this external source of excitation was necessary for activity in the network. In our simulations, each unit was fed with a random input, uncorrelated across units, but correlated in time:

$$\tau_N \dot{h}(\Theta, t) = -h(\Theta, t) + \eta(\mu, \sigma) \qquad (12)$$

where $\eta$ was a normal distributed random variable with mean $\mu$ and s.d. $\sigma$. This random background input could be combined with an additional direction selective component, restricted to a sub-set of the units, centred around a given direction,

$$h(\Theta, t) = \beta \exp\left( -(\Theta - \varphi)^2 / 2\kappa^2 \right) \qquad (13)$$

where $\beta$ controlled the strength of this component (with respect to the background one) and $\kappa$ regulated the degree of selectivity around the central selected direction $\varphi$ (cf. Supplementary Table 4 for network parameters values).

The degree of concentration of the pyramidal cell activity was measured as the activity bump coherence,

$$\Omega = \frac{\left| \sum\limits_{k=1}^{N_{\text{Pyr}}} e^{i\Theta_k} r_E(\Theta_k) \right|}{\sum\limits_{k=1}^{N_{\text{Pyr}}} r_E(\Theta_k)} \qquad (14)$$

Simulations consisted in integrating the network dynamics during 200 time steps of 1 ms. When only white noise was fed into the system, the simulation consisted in reproducing network dynamics starting from a random activity configuration. When studying the effect of directionally selective external inputs, the network received random and selective external inputs to the pyramidal cell layer for the first 60 time steps with the latter then gradually fading away between time steps 60 and 80. This procedure was repeated with the directional input sequentially centred over each of the cells preferred directions.

**Data availability.** The data that support the findings of this study are available from the corresponding authors upon reasonable request. All simulations were performed using MATLAB custom code, available from the authors.

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

## Acknowledgements

This work was supported by the Région Ile-de-France and Fondation pour la recherché Medicale (J.S.) and by ANR Grant JCJC R10206DD (D.F.). The research leading to these results also benefitted from the programme 'Investissements d'avenir' ANR-10-IAIHU-06, and we gratefully acknowledge financial support from the ERC (322721, R.M.). We thank Karl Deisseroth for making available AAV-hSyn-hChR2(H134R)-EYFP. We thank Dominique Debanne for discussions, and Matthew Nolan and Bruno Delord for comments on the manuscript. *In vivo* spike-train data were collected in the lab of Edvard and May-Britt Moser (Trondheim, Norway).

## Author contributions

J.S. and D.F. conceived and designed research, interpreted the data and supervised the experiments. All *in vitro* recordings were carried out and analysed by J.S., with the help of M.N. for recordings with optical stimulations and experiments on parvalbumin interneurons. B.M., M.N. and J.S. performed stereotaxic viral injections. I.C. provided tools for analysis. F.S. implemented the model and run the simulations. C.N.B. performed *in vivo* recordings and co-coordinated the study. J.S., F.S., C.N.B., R.M. and D.F. wrote the paper.

## Additional information

**Competing interests:** The authors declare no competing financial interests.

