## [Peer Review File · Nature Communications]

Reviewers' comments:

Reviewer #1 (Remarks to the Author):

In this study Simmonet and colleagues analyzed the connectivity of the ATN input to the presubiculum and the local microcircuit within the presubiculum in acute slices, and in conjunction with in vivo data and modeling they propose a scenario according to which this circuit may support the specific tuning of head direction neurons in the presubiculum.

The authors present elegant and convincing in vitro and in vivo data and demonstrate that Martinotti cells and pyramidal cells in layer III form a feedback inhibitory circuit. I congratulate the authors for a study that is original, important for the field and that sets the ground for future interesting work that will rely on these data. It was a great pleasure to read this manuscript that contains all information needed to make a strong case for the proposed hypothesis. The topic is well introduced and the discussion is beautifully written. The authors point out to similarities of Martinotti cell synapses in other cortical areas and those in the presubiculum, but also to differences that merit further investigations. Of course, this study is complete as is, but future studies may address some of the interesting points raised in the discussion. The study will be most welcome also for researchers studying the medial entorhinal cortex as some of the findings are likely to have some bearings on that brain region as well.

My suggestions are all minor and do not require further experiments. The results section would profit from small suggested changes that would make it easier for the reader to follow the arguments of this really beautiful study.

The authors write that ATN fibers... "did not project ..., to the adjacent subiculum or entorhinal cortex (Fig. 2a,b)" (Page 5). In reality Fig. 2b seems to show some faint signal in the parasubiculum as if some axons project there. The authors should show a higher magnification of the area for better evaluation.

Last sentence second subchapter results: "We examined this recurrent inhibitory control of thalamic input to pyramidal cells in dual patch clamp records from presubicular PCs and MCs in X98-SST mice (n = 54)." makes no sense here and it is not clear if n=54 mice or cells.

Page 6: "For single spikes, the transfer rate from pyramidal cells to Martinotti cells was very low, 0.12 ± 0.02 (median = 0.08; n = 44, Fig. 3e,f)." Fig. 3f does not exist (see also lower in the paragraph). In this subchapter efficiency and potency from PC to MC are negative numbers when expressed in pA. In the following subchapters and in the supplementary table 2 they are expressed as absolute numbers (positive). Maybe the authors indicate at the beginning that they will use throughout the manuscript absolute

values, and modify the graph diagrams and bars accordingly.

Please use the correct formatting when referring to the figures (i.e. Fig. 3c,d and not Fig. 3C,D).

It is not really clear how cumulative efficacy was calculated for Fig. 4f and 5f. A better description would be helpful.

I suggest that the term 'unmuting' should be defined when used the first time since it is not a common electrophysiological term. Otherwise one wonders whether it refers to a specific protocol to unsilence some connections (a tetanus for example), to unsilencing putative silent synapses or to a phenomenon that eventually increases presynaptic release.

Page 8, In vivo recordings were performed in rats. The authors used spike trains from isolated head direction units within their preferred range in vivo as depolarizing current commands to pyramidal cells in mouse slices. This is no problem for me but it should be explicitly stated in the text.

Page 9: Fig. 8h and Fig. 8i do not exist (they are cited several times on this page).

The model part: for the "non expert" I would explicitly state somewhere that the attractor network model does not employ recurrent excitation between pyramidal cells.

Figures, general comment: It is difficult to read the figures when for negative values expressing higher current flow zero is the highest value and the -max (-20 for Fig. 3d for example) is the lowest. It makes comparisons difficult. See for example Fig. 2f. For a 2D diagram expressing a linear fit (Fig. 1e where zero efficacy must correspond to zero average amplitude) this scale is misleading.

Figure 2b: higher magnification of parasubiculum and presubiculum to visualize axons is warranted; Fig 2f and g: why not use pA instead of nC?

Supplementary Fig1 (d-e) legend: "(d-e) These two methods were applied for $n = 11$ MC-to-PC synapses (b) and $n = 17$ PC-to-MC synapses (c) to test their agreement. Efficacy and Average amplitude were highly correlated for both synapses." (b) refers to (d) and (c) refers to (e) I guess.

Supplementary Fig. 3c please indicate time scale bar.

Reviewer #2 (Remarks to the Author):

The manuscript by Simonnet et al. reports investigations into the role of feedback inhibition provided by Martinotti cells (MCs) in processing head direction information in the rodent presubiculum (PS). The authors use intracellular recordings from pyramidal cells (PC) and

MCs in acute PS slice in vitro and computational modeling to find that (1) thalamic afferents preferentially innervate PCs, (2) MCs receive facilitating excitatory synaptic inputs from PCs, (3) and a network model incorporating reciprocal PC-MC connections with facilitating excitation generates and maintains attractor dynamics with directionality. Together, the authors conclude that this inhibitory circuit motifs is tuned to refine and maintain head direction information in the PS.

How head direction information is generated and processed in the brain at cellular and microcircuit level is a question of great importance. In general, the relevance of feedback inhibition by MCs for refining head direction firing of PCs is an interesting topic. Finally, assigning a potential role of the well-known facilitating short-term synaptic dynamics of PC-to-MC in this process is intriguing. While the in vitro slice physiology in the manuscript is of high quality, the conceptual advancement of the study is largely limited to re-cataloguing of otherwise well-known information about synaptic physiology and dynamics at this canonical feedback circuit motif. Because of these limitations, together with technical shortcomings, the paper is better suited to a more specialized journal and audience.

Some of these major concerns are detailed below.

1) A major shortcoming of the study is that the authors only look at the consequence of MC inhibition on action potential generation in PCs induced by somatic current injections, while they completely ignore the fact that MCs target and inhibit distal dendrites of PCs. There is a wealth of evidence across various cortical and hippocampal circuits (e.g. Murayama, *Nature*, 2009; Lovett-Barron et al., *Nature Neuroscience*, 2012), including also classic works by one of the senior authors on the current manuscript (Miles et al., *Neuron*, 1996), which demonstrate that the primary role of the dendritic inhibition is to regulate local integration of excitatory inputs and dendritic electrogenesis. The exclusive focus of the current study on the effects of dendritic inhibition originating from MC on somatic action potential generation is a critical limitation of the study design as it largely limits the conclusions one can make on the complex nature of interactions between dendritic excitation and MC-inhibition that may take place in dendrites of PCs. On a related note, the authors use low-Chloride intracellular solution when record from PCs in order to render MC- GABA-A inhibition hyperpolarizing and therefore more detectable/effective at the soma, but why would this small and filtered somatic hyperpolarization be the major means by which MCs influence their postsynaptic targets?

2) The major part of the study is in vitro synaptic physiology detailing properties of short-term facilitation at PC-to-MC synapses and the consequential recurrent inhibition from MCs to PCs. While this part is of high quality and the analysis is convincing, the findings in the most part aren't surprising, other than reassuringly demonstrating that MCs in PS also receive recurrent excitatory synapses with robust facilitation, similar to what has been already widely and consistently reported across many other cortical regions (e.g. Gulyas et al., *Nature*, 1993; Ali and Thomson, *J Physiol.*, 1998; Silberberg and Markram, *Neuron*, 2007).

3) In general, the authors' implementation of recurrent inhibition with short-term facilitation in a network model to generate attractor dynamics with directionality is interesting. However, it is not really novel, as the consequences of short-term facilitation in general, and MC recurrent inhibition with facilitatory STP on attractor dynamics in particular, have already been modeled quite extensively (e.g. Itskov et al., *Front. Comput. Neurosci.*, 2011; Krishnamurthy et al., *PLoS One*, 2012; Fung et al., *Neural Comput.*, 2012). The authors did not seem to acknowledge these previous modeling efforts. Furthermore, many parts of the model implementation are unclear: e.g. (1) is 'pruning' a consequence of STP dynamics, or it is a prerequisite to obtain/refine head direction firing, (2) even the somewhat more interesting findings with in vitro physiology (e.g. the meta-plasticity - 'unmuting' of PC-to-MC synapses) are not implemented in the model.

4) At the end of the day, it remains unknown what the actual influence of MC recurrent inhibition is on head direction firing of PCs in the behaving animal. The "in vivo" part of the study appears to be limited to a single sample in vivo trace (Fig 7a,b) which is used to constrain the in vitro spiking pattern of PCs. Rather, in vivo manipulations of defined interneuron classes with simultaneous read out of changes in tuning properties of principal neurons, as it has been performed many times in other cortical circuits, would have been appropriate and informative approaches here. On a related note, the authors acknowledge the Mosers' lab for the in vivo spike train data, while the authors also describe in vivo electrophysiology in great details as if it was an integral part of their study. Given that even this single in vivo sample may have been actually obtained from an outside lab, it is an over-exaggeration that the word 'in vivo' is mentioned at least dozen times throughout the study.

Reviewer #3 (Remarks to the Author):

This paper describes the microcircuitry of the postsubicular head-direction cell network, and proposes a connectivity model for the formation and maintenance of stable directional outputs that are inherited from the thalamus. The functional connectivity from thalamus to the presubiculum is delineated by optically exciting neurons in thalamus and recording from Martinotti interneurons and pyramidal neurons in presubiculum. These experiments demonstrate that the innervation of pyramidal cells from thalamus is strong and monosynaptic, while the excitation of Martinotti cells is driven by recurrent feedback from the PCs in subiculum. Cell pairing experiments show that the PCs and MCs are richly connected, while PCs are seldom connected to each other. The authors then used in vivo recordings of presubicular HDCs as training stimuli for in vitro connected MC-PC pairs, demonstrating that the fast bursting seen in vivo is sufficient to drive (or unmute) PC to MC communication. Finally, by varying the timing of the activity of the MC IPSP, the authors provide evidence for a network arrangement highly suggestive of a center-surround inhibition attractor network between PCs and MCs. A two-layer ring attractor network model can adequately replicate the observed characteristics of the PC-MC pairing in vivo and in vitro.

I found the experiments to be well-executed and the implications carefully thought out. I

believe the findings provide important new findings regarding the cellular and synaptic basis of attractor networks capable of generating head direction signals in the presubiculum. However, I have a few comments:

1. It would be nice to see more comprehensive images of EC, especially layers 3 and 5, in order to assure that ATN neurons do not innervate the HD cells contained here. This distinction (serial transfer from ATN to Subiculum to EC vs ATN to both Subiculum and EC in parallel) has important functional implications for the genesis and role of the HD signal in EC.

2. It's not clear why such a wide volume window (50-150nl) of virus was injected into the ATN. Some justification of this would be welcome. It would also be useful to quantify the extent of infection of ATN and to determine whether any extra-regional overlap was produced by the injections, perhaps in a separate figure.

3. While not critical, the use of different species for in vivo experiments compared to in vitro experiments is troubling. While mice are obviously preferred for in vitro experiments due to genetic access and rats for in vivo experiments due to the higher channel counts and unit yield, there are important differences in the organization of several microcircuits between these species (the distribution of cells throughout entorhinal cortex seems to be markedly more diffuse in mice vis-à-vis rats, for example). Some justification for the switch between these species should be made, with these caveats made more explicit.

4. Various comparisons are made under the section titled 'electrophysiology of presubicular Martinotti and pyramidal cells' but no statistics are presented. In addition, p values should be presented in table 1 for comparisons that are presented as 'different' in the text.

5. In Figure 1d, it's not clear to me what Mann-Whitney value is being reported. In addition, it seems an important comparison is missing; namely the comparison of 25 ms charge for MC in standard versus TTX-AP5 solutions. It appears as if the response of MC is very small even in the standard ACSF and it's not clear this is significantly different in the TTX-AP5 solution.

6. It would be helpful if justification for why the authors use X98-SST mice are used is presented when they are first mentioned (top of page 6).

7. Figure legend 1e: not sure what is meant by "separates of PCs"

8. Figure 4a: what is the difference between the traces on the left and right?

9. Figure 4g: what are the small red dots near the value 3? Are these individual data points (in black)?

10. Page 4: should be "presubicular [recordings] in vivo"

11. Page 5: not sure what is meant by "layer 1 as for other neocortical Martinotti cells"

12. Page 9: should be "in [recordings] from unidirectionally connected"

Reviewer #4 (Remarks to the Author):

In the manuscripts titled, 'Activity dependent feedback inhibition supports head direction coding in the presubiculum' Simonnet and colleagues identify and characterize inputs to the pyramidal cell (PC) and somatostatin positive cells (SOM) of the presubiculum in mice slices. The manuscript presents numerous findings that center around the theme of identifying the circuit properties of the presubiculum. First, the authors show that between PC and SOM cells, presubicular inputs from the anterodorsal nucleus of the thalamus only synapse onto pyramidal cells. The authors demonstrate that the PC and SOM cells form a densely interconnected network including a high probability of reciprocal connections. The authors show that the SOM-to-PC synapse efficiency has minimal activity dependence while the PC-to-SOM connection is highly activity dependent. The PC-to-SOM synapse efficacy increases with increasing PC activity, effectively unmuting the connection. The authors then compared reciprocally versus non-reciprocally connected SOM/ PC pairs for differences in inhibitory dynamics. The authors claim that their data demonstrates that feedback inhibition from a reciprocally connected SOM cell has minimal influence while non-reciprocally connected SOM cells have increased inhibitory influence on the PC. Finally, the authors present a computational model of the head direction circuit from which they conclude that the SOM cells support head direction coding in the presubiculum.

First and foremost, this is an outstanding manuscript both in terms of quality of research presented as well as in likely importance for the community. With regard to importance, the head direction circuit is a key component of the neural circuitry underlying the "brain's GPS" - a circuit celebrated by the 2014 Nobel Prize. The current work provides a vivid and much needed depiction of how the head direction signal is supported by the presubiculum, a key step in the relay of the head direction signal. This manuscript provides diverse, yet thematic, data regarding the functional structure of pyramidal and somatostatin positive interneurons circuitry in the presubiculum. This data answers open questions regarding how thalamic input is integrated by the presubiculum and makes substantial progress in building a complete picture of presubicular circuitry. With regard to the research, the questions posed are well formed and, in most cases, the empirical data provide conclusive answers to the questions.

I am highly enthusiastic about this manuscript and its potential to make a significant contribution to the field. However, there are several points that require attention prior to publication. I have described these as 'major points' below. Despite the label 'major' it should be noted that addressing the 'major points' need not take extensive work or time to address - I have indicated what I see to be the passable easy fix for each along with the not-so-easy fix. I have also included 'minor points' for the authors consideration for improved clarity / value of the manuscript.

Major points

Most critically, the model included in the manuscript strays too far from the current data. It misrepresents the data and introduces the distinct possibility of leading to false conclusions. By my understanding, the model architecture and synaptic dynamics both deviate from data reported in the manuscript in functionally consequential ways that prevent conclusions from being drawn from the model regarding the empirical data. With regard to synaptic dynamics, the model does not explicitly implement the reported activity dependent facilitation at PC to SOM cell synapses. Instead, it uses a winner take all mechanism which is not an acceptable approximation for the actual dynamics of the facilitating synapse. Additionally, the empirical data shows the latency of inhibitory feedback changes inhibitory efficacy but rather than allow the model to demonstrate this, the model design hard codes them into the simulation. Because they are hard coded, the model behavior cannot be used to draw a conclusion regarding the functional consequences of the latency data. With regard to architecture, a minor issue is that the connectivity ratios observed in the data are not literally replicated in the model and additional strong assumptions are made regarding how recurrent connections (called 'strong connections') fit into the architecture. On both fronts, there is no analysis as to whether these assumptions are required for the function of the model. Generally speaking, I'm in favor of including a model, especially because the data virtually beg for one. However, when I say this, I mean that the model should take replicating the observed data as its top priority and, from that standpoint, demonstrate the functional consequence of the observed phenomena. The current model uses abstractions of the current data (that I believe to be poor abstractions) which then weakens or eliminates the ability of the model to conclusively demonstrate that observed physiology has the functional properties cited by the authors. This is a significant flaw of the paper that must be addressed. Easy fix - drop the model, the paper does not need it to serve as a landmark work. Omitting the model may even spur a flurry of modeling work by others that will draw further attention to this data. Not-so-easy fix - redo the model to implement high fidelity realizations of the empirical data and from there demonstrate the functional consequences.

The title, abstract, and discussion say that feedback inhibition supports head direction coding but the current data does not technically show this. It may be that the authors conclude this from the modeling work, however as noted above, I'm completely unconvinced by the model in its present form. The empirical data does not specifically show anything about head direction coding because it is done in slice. Easy fix: update the wording title, abstract, and discussion. My glowing enthusiasm of this manuscript is despite these statements not because of them. Dropping them will not decrease my excitement for seeing this manuscript in press. Not-so-easy fix: inactivate presubicular SOM cells in vivo while recording head direction cells, show that this reduces directional coding, and then conclude that local SOM cells support head direction coding.

The analysis of recurrent versus non-recurrent inhibition does not conclusively demonstrate that there are functional differences in inhibitory dynamics between these two types of

inhibition. Rather, the data show a difference in low-latency ($< 10\text{ms}$) IPSPs versus non-low-latency IPSPs ($> 10\text{ms}$). No data is shown regarding a difference in IPSP latency between recurrent and non-recurrent cell pairs. This data is needed to connect these functional differences resulting from latency to the connectivity differences. Such an analysis should compare the observed latency distributions to that expected by chance assuming that the PC cell has no influence on the SOM cell activity. This analysis would make it possible to identify the IPSP latencies that occur in recurrent pairs that occur more often than chance and, with this knowledge, consider the influence of IPSPs that arrive within this window. Similarly, this analysis could show if the distribution of IPSP latencies observed in non-recurrent pairs differs from what would be expected by chance.

The authors report elsewhere (Nassar et al., 2015) that there is not a 1-to-1 mapping between the SOM label and Martinotti cells but the present work uses the name Martinotti cell for the fluorescently labeled SOM cells. Although the present paper notes that biocytin fills revealed Martinotti morphologies, no indication is given that analyses were restricted to those confirmed to have this morphology.

Regarding stats:

Some indication that the included values in a statement like "gain was lower in PCs ($0.373 \pm 0.016 \text{ Hz.pA}^{-1}$)" indicate mean and standard deviation should be given.

Additional attention should be given to indicating the n that goes into the included stats (e.g., n is not indicated in the first paragraph of the results or in the section titled 'Stable Martinotti cell inhibition. . .').

Is $n=4$ sufficient to reliably estimate the decay rate of the PC-to-SOM synapse facilitation?

Minor points

It would be powerful to include a putative mechanism for the facilitation. Is it presynaptic? Postsynaptic? Is it related to the PC to SOM activity dependent facilitation previously shown to be dependent on presynaptic NMDA receptors by Buchanan et al., (2012, Neuron)?

A comparison of the number of reciprocal connections observed to the number of connections expected given the probability of SOM- \rightarrow PC and PC- \rightarrow SOM would be welcomed (I boarder on saying should be obligatory). This should include statistics indicating if it is significantly greater than expected by chance.

Some indication of which subset of cell pairs were examined in subsequent analyses should be included (e.g., the one described in the section titled 'Stable Martinotti cell inhibition. . . ' or in section titled 'Repetitive stimulation unmutes. . . ').

The manuscript shows a 7-11s decay rate of the PC-to-SOM synapse - a discussion or,

better yet, analysis of how this relates to the time scale of behavior would be welcomed. For example, what is the typical latency that an animal revisits a given head direction and what is the implication of this for the instantaneous distribution of how strongly facilitated the PC-to-SOM synapses are over the network? (This would be an exemplar analysis to do with the high fidelity variant of the model I argue for above.)

There is a jarring disconnect between the introduction and the first paragraph of the results. A sentence or two at the outset of this section motivating that these two cell types were of particular focus and that the first step was to characterize their respective basic properties may help make this less disorienting.

The sudden appearance of PV+ cell analysis in the section titled 'How in vivo head direction signaling. . .' suddenly made me wonder why they had been ignored otherwise. Some improved introduction may mitigate this. In the discussion, I would welcome some time spent on incorporating how the PV+ cells might fit into all that has been shown in the paper.

We would like to thank the reviewers for helpful suggestions on how our ms could be improved. We have performed new experiments in vivo and slice, made further analyses and extended our network model to address the points raised (see changes to Fig. 6, Fig. 9, new Supplementary Fig. 1 and Supplementary Fig. S5). As suggested we modify the title, abstract, and discussion. The title of our ms is now "How activity dependent feedback inhibition may maintain head direction signals in mouse presubiculum".

Changes have been integrated into the manuscript and highlighted. Please see below for specific details of the revisions. We hope that the reviewers will be satisfied with the changes we made and that our ms is now acceptable for publication in Nature Communications.

Reviewers' comments:

Reviewer #1 (Remarks to the Author):

In this study Simonnet and colleagues analyzed the connectivity of the ATN input to the presubiculum and the local microcircuit within the presubiculum in acute slices, and in conjunction with in vivo data and modeling they propose a scenario according to which this circuit may support the specific tuning of head direction neurons in the presubiculum.

The authors present elegant and convincing in vitro and in vivo data and demonstrate that Martinotti cells and pyramidal cells in layer III form a feedback inhibitory circuit. I congratulate the authors for a study that is original, important for the field and that sets the ground for future interesting work that will rely on these data. It was a great pleasure to read this manuscript that contains all information needed to make a strong case for the proposed hypothesis. The topic is well introduced and the discussion is beautifully written. The authors point out to similarities of Martinotti cell synapses in other cortical areas and those in the presubiculum, but also to differences that merit further investigations. Of course, this study is complete as is, but future studies may address some of the interesting points raised in the discussion. The study will be most welcome also for researchers studying the medial entorhinal cortex as some of the findings are likely to have some bearings on that brain region as well. My suggestions are all minor and do not require further experiments. The results section would profit from small suggested changes that would make it easier for the reader to follow the arguments of this really beautiful study.

We thank the reviewer for the positive evaluation.

The authors write that ATN fibers... "did not project ..., to the adjacent subiculum or entorhinal cortex (Fig. 2a,b)" (Page 5). In reality Fig. 2b seems to show some faint signal in the parasubiculum as if some axons project there. The authors should show a higher magnification of the area for better evaluation.

We add higher magnification 20x images obtained with a Zeiss confocal LSM710 of the area as a new Supplementary Figure 1. They show presubiculum, parasubiculum, subiculum and entorhinal cortex (cf. also reviewer 3). While there is massive thalamic innervation precisely in the superficial layers of dorsal presubiculum, we also note a faint signal indicating that a few thalamic axons run into parasubiculum and entorhinal cortex. We change the text (Page 5) to read "A few axons were present in deep layers and in parasubiculum. The zone of thalamic innervation ended abruptly at the border to the adjacent subiculum. Very few axons if any were present in entorhinal cortex (Fig. 2a, b and Supplementary Fig. 1)".

Last sentence second subchapter results: "We examined this recurrent inhibitory control of thalamic input to pyramidal cells in dual patch clamp records from presubicular PCs and MCs in X98-SST mice (n = 54)." makes no sense here and it is not clear if n=54 mice or cells.

We changed this sentence to read: "In return, Martinotti cells may provide a recurrent inhibitory control of pyramidal cells. We next examined the presubicular PC-MC connectivity pattern and its dynamics using dual patch clamp records in X98-SST mice."

Connectivity was tested in paired recordings from slices of 55 X98-SST mice. This is now indicated in the legend of Fig. 3a. (Data on 4 paired recordings from 1 additional animal, performed to revise Fig. 6, are now included, and connectivity statistics are updated.)

Page 6: "For single spikes, the transfer rate from pyramidal cells to Martinotti cells was very low, 0.12 {plus minus} 0.02 (median = 0.08; n = 44, Fig. 3e,f)." Fig. 3f does not exist (see also lower in the paragraph). In this subchapter efficiency and potency from PC to MC are negative numbers when expressed in pA. In the following subchapters and in the supplementary table 2 they are expressed as absolute numbers (positive). Maybe the authors indicate at the beginning that they will use throughout the manuscript absolute values, and modify the graph diagrams and bars accordingly.

We now give absolute numbers in all figures, tables and in the text. The reference to Fig. 3f is removed.

Please use the correct formatting when referring to the figures (i.e. Fig. 3c,d and not Fig. 3C,D).
This point is now corrected.

It is not really clear how cumulative efficacy was calculated for Fig. 4f and 5f. A better description would be helpful.

We add a sentence of clarification in the methods section: "Cumulative efficacy was calculated as the sum of efficacy over time and provides a temporal dynamic". In the figure legends we add "summed efficacy over time".

I suggest that the term 'unmuting' should be defined when used the first time since it is not a common electrophysiological term. Otherwise one wonders whether it refers to a specific protocol to unsilence some connections (a tetanus for example), to unsilencing putative silent synapses or to a phenomenon that eventually increases presynaptic release.

We use the term "unmuting" to indicate an extreme form of facilitation, where the synapses are initially almost silent, but EPSPs are elicited more efficiently when pyramidal cells fire at high frequency (cf. Fig. 3d and Fig. 5 of this ms and Fig. 6 of Silberberg & Markram, 2007). We add an explanatory sentence in the text (bottom of page 7).

Page 8, In vivo recordings were performed in rats. The authors used spike trains from isolated head direction units within their preferred range in vivo as depolarizing current commands to pyramidal cells in mouse slices. This is no problem for me but it should be explicitly stated in the text.

We change the text to: "from rats running in an open field" (page 8).

Page 9: Fig. 8h and Fig. 8i do not exist (they are cited several times on this page).

We have corrected this.

The model part: for the "non expert" I would explicitly state somewhere that the attractor network model does not employ recurrent excitation between pyramidal cells.

We now state it explicitly (page 10): "These models mostly rely on strong excitatory connections between cells with similar preferred directions. We asked whether a model based on experimentally measured connectivity, strength and dynamic behavior of synapses in recurrent Martinotti-cell circuits could

generate attractor network dynamics, in the absence of recurrent excitation between pyramidal cells.”

Figures, general comment: It is difficult to read the figures when for negative values expressing higher current flow zero is the highest value and the -max (-20 for Fig. 3d for example) is the lowest. It makes comparisons difficult. See for example Fig. 2f. For a 2D diagram expressing a linear fit (Fig. 1e where zero efficacy must correspond to zero average amplitude) this scale is misleading.

We thank the reviewer for his comment. We have modified figures according to the suggestion.

Figure 2b: higher magnification of parasubiculum and presubiculum to visualize axons is warranted; Fig 2f and g: why not use pA instead of nC?

As stated above, we add 20x images as a Supplementary Fig. 1. We also provide a 60x image of layer 3 of presubiculum and of entorhinal cortex layer 3 and 5.

Fig 2f and g: We chose to measure charge transfer during the first 25 ms (the area under the PSC), rather than amplitude, since events sometimes comprised multiple peaks.

Supplementary Fig1 (d-e) legend: "(d-e) These two methods were applied for n = 11 MC-to-PC synapses (b) and n = 17 PC-to-MC synapses (c) to test their agreement. Efficacy and Average amplitude were highly correlated for both synapses." (b) refers to (d) and (c) refers to (e) I guess.

We have changed the legend accordingly.

Supplementary Fig. 3c please indicate time scale bar.

We add a time scale bar. We also correct a mistake in the vertical scale in panel c.

Reviewer #2 (Remarks to the Author):

The manuscript by Simonnet et al. reports investigations into the role of feedback inhibition provided by Martinotti cells (MCs) in processing head direction information in the rodent presubiculum (PS). The authors use intracellular recordings from pyramidal cells (PC) and MCs in acute PS slice in vitro and computational modeling to find that (1) thalamic afferents preferentially innervate PCs, (2) MCs receive facilitating excitatory synaptic inputs from PCs, (3) and a network model incorporating reciprocal PC-MC connections with facilitating excitation generates and maintains attractor dynamics with directionality. Together, the authors conclude that this inhibitory circuit motifs is tuned to refine and maintain head direction information in the PS.

How head direction information is generated and processed in the brain at cellular and microcircuit level is a question of great importance. In general, the relevance of feedback inhibition by MCs for refining head direction firing of PCs is an interesting topic. Finally, assigning a potential role of the well-known facilitating short-term synaptic dynamics of PC-to-MC in this process is intriguing. While the in vitro slice physiology in the manuscript is of high quality, the conceptual advancement of the study is largely limited to re-cataloguing of otherwise well-known information about synaptic physiology and dynamics at this canonical feedback circuit motif. Because of these limitations, together with technical shortcomings, the paper is better suited to a more specialized journal and audience.

We agree with the reviewer that the generation of head direction information is of great interest.

We disagree, respectfully, that events contributing to this process in the presubiculum can already be explained by ‘well-known information ... about ... this canonical feedback circuit’.

We note the presubiculum is not a typical neocortical area, but rather part of the transitional periarchicortex. The cytoarchitecture and circuit function differs from both neocortex and hippocampus. We now refer to the view from Berger et al. (2009) that microcircuit connectivity and dynamics of specific

pathways “might reflect or define the function of a cortical area”.

In support of functional differences we note that head direction cells typically sustain firing frequencies with little adaptation (Taube and Muller, 1998). In many, maybe not all, other cortical areas principal cells respond to a stimulus with only transient activity. For instance neurons of barrel cortex respond largely at the onset of whisker stimulation (Gabernet et al., 2005). We suggest then that the ‘canonical’ nature of circuit behavior needs to be examined with care.

Our data certainly shows that synapses which excite SST interneurons facilitate as has been found in previous work (eg Silberberg & Markram). We suggest that basal transmission is more depressed and the degree of facilitation during repetitive activation is greater than at similar connections in other regions (Ali & Thomson 1998; Silberberg & Markram, 2007)

Some of these major concerns are detailed below.

1) A major shortcoming of the study is that the authors only look at the consequence of MC inhibition on action potential generation in PCs induced by somatic current injections, while they completely ignore the fact that MCs target and inhibit distal dendrites of PCs. There is a wealth of evidence across various cortical and hippocampal circuits (e.g. Murayama, Nature, 2009; Lovett-Barron et al., Nature Neuroscience, 2012), including also classic works by one of the senior authors on the current manuscript (Miles et al., Neuron, 1996), which demonstrate that the primary role of the dendritic inhibition is to regulate local integration of excitatory inputs and dendritic electrogenesis. The exclusive focus of the current study on the effects of dendritic inhibition originating from MC on somatic action potential generation is a critical limitation of the study design as it largely limits the conclusions one can make on the complex nature of interactions between dendritic excitation and MC-inhibition that may take place in dendrites of PCs. On a related note, the authors use low-Chloride intracellular solution when record from PCs in order to render MC- GABA-A inhibition hyperpolarizing and therefore more detectable/effective at the soma, but why would this small and filtered somatic hyperpolarization be the major means by which MCs influence their postsynaptic targets?

Of course we agree. We have attempted to improve our discussion on this point to respond to this concern. Dendritic inhibition mediated by MCs acts to counter the effects of local, dendritic EPSPs. But also by suppressing dendritic electrogenesis, it seems likely to affect action potentials propagating to the soma. We note that presubicular cells that signal head direction are identified by their firing. We have modified the discussion (page 14) to suggest that MC mediated inhibition may affect firing as a dendritic hyperpolarization propagates and/or by opposing dendritic EPSPs.

2) The major part of the study is in vitro synaptic physiology detailing properties of short-term facilitation at PC-to-MC synapses and the consequential recurrent inhibition from MCs to PCs. While this part is of high quality and the analysis is convincing, the findings in the most part aren't surprising, other than reassuringly demonstrating that MCs in PS also receive recurrent excitatory synapses with robust facilitation, similar to what has been already widely and consistently reported across many other cortical regions (e.g. Gulyas et al., Nature, 1993; Ali and Thomson, J Physiol., 1998; Silberberg and Markram, Neuron, 2007).

As reviewer #1 states, our results “point to similarities of Martinotti cell synapses in other cortical areas and those in the presubiculum, but also to differences...”. We add material on this point to the discussion (page 13). Specifically, our data shows that the transfer rate at the presubicular PC-to-MC synapse is very low, and that “even after unmuting... it remains quite low ..., compared to responses elicited by similar stimuli at neocortical PC-to-MC synapses in layer 3 (Faselow et al., 2008) or layer 5 (Silberberg & Markram 2007)” (cf. discussion, p. 13). Synaptic facilitation in the presubiculum differs in that kinetics are very slow, possibly an adaptation for the persistent discharges of head direction cells. High frequency sustained firing is needed to unmute the excitation of SST interneurons. For comparison, a 3-fold increase in average EPSP amplitude was obtained after 8 stimuli at 20 Hz in somatosensory cortex (Faselow et al., 2008), while our data from presubiculum show 30 stimuli at 30 Hz were needed to induce a similar degree

of facilitation. These differences are significant and we feel adapted to the processing that occurs in the presubiculum.

3) In general, the authors' implementation of recurrent inhibition with short-term facilitation in a network model to generate attractor dynamics with directionality is interesting. However, it is not really novel, as the consequences of short-term facilitation in general, and MC recurrent inhibition with facilitatory STP on attractor dynamics in particular, have already been modeled quite extensively (e.g. Itskov et al., Front. Comput. Neurosci., 2011; Krishnamurthy et al., PLoS One, 2012; Fung et al., Neural Comput., 2012). The authors did not seem to acknowledge these previous modeling efforts. Furthermore, many parts of the model implementation are unclear: e.g. (1) is 'pruning' a consequence of STP dynamics, or it is a prerequisite to obtain/refine head direction firing, (2) even the somewhat more interesting findings with in vitro physiology (e.g. the meta-plasticity - 'unmuting' of PC-to-MC synapses) are not implemented in the model.

We have modified the network model according to comments of reviewers #2 and #4. Facilitation is now implemented explicitly and 'unmuting' of the excitatory synapse is modeled as an extreme form of facilitation. We acknowledge previous models that have suggested how synaptic facilitation might stabilize the network (although in these works the interaction between excitatory and inhibitory cells is not explicitly represented, as they all deal with effective connectivity between excitatory cells), adding references to Itskov et al. (2011), Krishnamurthy et al. (2012) and Fung et al. (2012) in the methods (p.20) and the discussion (p. 14). We also add a paragraph to the methods (p. 18-19) to clarify the novelty of the model concept, and stressing our intention to study the behavior of interneurons in a model for head direction selectivity, by directly including them in the network structure. In this respect both points raised by the reviewer are relative to the way the layer of Martinotti cells is integrated in the modeled presubiculum network.

Point (1), the 'pruning' is part of the network architecture. It concerns the dependency of inhibitory connections on the relative preferred phase of pyramidal units and in particular the absence (or strong reduction) of mutual inhibition between pyramidal units with similar phase preference. We explain in the methods (p. 19), how this property is required for building network models that generate attractor dynamics with directionality as the pattern of connectivity contains the information about which of the pyramidal units are supposed to fire together and which are not. (ref. 20-23).

Point (2) is now modeled as suggested. SST-type interneurons are explicitly modeled, as are experimental data on the kinetics of their recruitment by facilitating synapses, following the suggestion of reviewers #2 and #4.

We feel these changes improve the model which is now both novel and more firmly based on experimental data. It lets us show that the activity bump of the network can exist even without facilitating Martinotti units, for some conditions (short persistence, large b_1 , cf. Supplementary Fig. S5b). In the absence of external inputs facilitating synapses are crucial to stabilize and maintain activity, well correlated with the initial externally imposed signal. Depressing synaptic dynamics always fail to maintain the activity coherence in time (modified Fig. 9e and Supplementary Fig. S5c).

So, while depressing synaptic dynamics are not completely incompatible with the generation of a bump of activity (modified Fig. 9d), they strongly interfere with the proper functioning of the continuous attractor network.

For testing the role of Martinotti cells in vivo (cf. next question below), the recording conditions should therefore be designed in a particular manner. It may not be sufficient to record in mice that are freely moving in an environment while silencing Martinotti cells. The experimental conditions should reduce any external input, such recordings in darkness (no visual input) or immobility (no angular velocity inputs from the inner ear), or during sleep.

4) At the end of the day, it remains unknown what the actual influence of MC recurrent inhibition is on head direction firing of PCs in the behaving animal. The "in vivo" part of the study appears to be limited to a single sample in vivo trace (Fig 7a, b) which is used to constrain the in vitro spiking pattern of PCs.

Rather, in vivo manipulations of defined interneuron classes with simultaneous read out of changes in tuning properties of principal neurons, as it has been performed many times in other cortical circuits, would have been appropriate and informative approaches here. On a related note, the authors acknowledge the Mosers' lab for the in vivo spike train data, while the authors also describe in vivo electrophysiology in great details as if it was an integral part of their study. Given that even this single in vivo sample may have been actually obtained from an outside lab, it is an over-exaggeration that the word 'in vivo' is mentioned at least dozen times throughout the study.

Charlotte Boccara obtained *in vivo* tetrode recordings in the Mosers' lab in Trondheim. This is stated in the acknowledgements, and we also briefly describe the methods, with a reference to Boccara et al., 2010. All slice work has been carried out in Paris.

How do SST-Martinotti cells affect the head direction firing of PC neurons in the presubiculum in freely moving animals? Surely the reviewer was trying to help us improve our ms, but we found the suggestion – *'in vivo manipulations of defined interneuron classes with simultaneous read out of changes in tuning properties of principal neurons'* was not precise. One interpretation is that we should record from interneurons of different classes (presumably SST+ and another maybe PV+) and from head direction sensitive pyramidal cells, manipulate interneuron firing and test effects of head direction signals. It's a lot to ask for 3 months from a lab that does not currently have the equipment and expertise for *in vivo* recordings. Furthermore we are not sure that such an approach would resolve the question.

[redacted]

[redacted]

Our empirical data do not demonstrate the mechanism of head direction coding *in vivo*. In order to avoid overstatement, we follow the suggestion of reviewer #4, and change the title of our ms to: “How activity dependent feedback inhibition may maintain head direction signals in mouse presubiculum”.

Reviewer #3 (Remarks to the Author):

*This paper describes the microcircuitry of the postsubicular head-direction cell network, and proposes a connectivity model for the formation and maintenance of stable directional outputs that are inherited from the thalamus. The functional connectivity from thalamus to the presubiculum is delineated by optically exciting neurons in thalamus and recording from Martinotti interneurons and pyramidal neurons in presubiculum. These experiments demonstrate that the innervation of pyramidal cells from thalamus is strong and monosynaptic, while the excitation of Martinotti cells is driven by recurrent feedback from the PCs in subiculum. Cell pairing experiments show that the PCs and MCs are richly connected, while PCs are seldom connected to each other. The authors then used *in vivo* recordings of presubicular HDCs as training stimuli for *in vitro* connected MC-PC pairs, demonstrating that the fast bursting seen *in vivo* is sufficient to drive (or unmute) PC to MC communication. Finally, by varying the timing of the activity of the MC IPSP, the authors provide evidence for a network arrangement highly suggestive of a center-surround inhibition attractor network between PCs and MCs. A two-layer ring attractor network model can adequately replicate the observed characteristics of the PC-MC pairing *in vivo* and *in vitro*.*

I found the experiments to be well-executed and the implications carefully thought out. I believe the findings provide important new findings regarding the cellular and synaptic basis of attractor networks capable of generating head direction signals in the presubiculum. However, I have a few comments:

1. It would be nice to see more comprehensive images of EC, especially layers 3 and 5, in order to assure that ATN neurons do not innervate the HD cells contained here. This distinction (serial transfer from ATN to Subiculum to EC vs ATN to both Subiculum and EC in parallel) has important functional implications for the genesis and role of the HD signal in EC.

We have added new high quality 20x laser confocal images of the parahippocampal region, in a new Supplementary Figure 1. They show thalamic axons ramify, at highest density, in the superficial layers of

presubiculum. A 60x magnification of EC layer 3 and 5, is also shown so that readers may judge an absence of thalamic axons. There is some faint green labeling, and as reviewer #1 suggests, it is hard to be certain that thalamic axons are totally absent. We therefore modify the text (Page 5): “A few axons were present in deep layers and in parasubiculum. The zone of thalamic innervation ended abruptly at the border to the adjacent subiculum. Very few axons were present in entorhinal cortex (Fig. 2a,b and Supplementary Fig. 1)”.

2. It's not clear why such a wide volume window (50-150nl) of virus was injected into the ATN. Some justification of this would be welcome. It would also be useful to quantify the extent of infection of ATN and to determine whether any extra-regional overlap was produced by the injections, perhaps in a separate figure.

Our viral injections were targeted to the anterior nuclei of the thalamus (ATN). Expression of ChR2-eYFP included the dorsal (AD) and ventral (AV) part of the ATN. Only mice with strong expression in ATN were included. All 5 mice had some expression in LD and in some cases a minor spillover to the reticular nucleus or to lateral habenular nucleus LHb, which we tolerated. From the literature it is clear that only AD, AV, LD project to the presubiculum. Connectivity data of the Allen Brain Atlas show no projection of the LHb to the hippocampal-parahippocampal area.

We chose injection volumes of 150 nl for 5 mice used in these experiments to be region-specific and also to cover the ATN. We now state this in the text (methods, page 15) and add Supplementary Fig. 1a. For larger injection volumes (up to 300nl), AAV spread was sometimes larger, but we still detected the same pattern of axon terminals in the presubiculum and only very faintly in MEC.

Our quantitative assessment of expression levels in thalamic and adjacent structures for the 5 mice used in this study is as follows:

mouse n°	AD	AV	LD	RT	LHb
#616	++++	++++	+	+	+
#617	++++	++++	++	-/+	-
#621	++++	+++	+	-/+	+
#622	+++++	+++++	++	+	+/-
#623	++++	+++	++	+	+

3. While not critical, the use of different species for in vivo experiments compared to in vitro experiments is troubling. While mice are obviously preferred for in vitro experiments due to genetic access and rats for in vivo experiments due to the higher channel counts and unit yield, there are important differences in the organization of several microcircuits between these species (the distribution of cells throughout entorhinal cortex seems to be markedly more diffuse in mice vis-à-vis rats, for example). Some justification for the switch between these species should be made, with these caveats made more explicit.

We agree with this point of course and modify the text accordingly (page 17).

4. Various comparisons are made under the section titled 'electrophysiology of presubicular Martinotti and pyramidal cells' but no statistics are presented. In addition, p values should be presented in table 1 for comparisons that are presented as 'different' in the text.

We have now added the statistics in the text and in Supplementary Table 1.

5. In Figure 1d, it's not clear to me what Mann-Whitney value is being reported. In addition, it seems an important comparison is missing; namely the comparison of 25 ms charge for MC in standard versus TTX-AP5 solutions. It appears as if the response of MC is very small even in the standard ACSF and it's not clear this is significantly different in the TTX-AP5 solution.

We correct this mistake in the legend for Figure 2d. In response to the concern we now use the Kruskal-Wallis + Dunn's post test to compare PC and MC responses, correcting for multiple comparisons. CTL condition suffices to show that MC cells are not recruited following thalamic activation. However, the PC response were very large, possibly due to a direct contact, or to local recurrent excitation. TTX-4AP let us check this point and also suppress recurrent excitation. 4AP was needed to increase axonal excitability to permit synaptic release. 4AP strongly alters many pre- and postsynaptic neuronal properties resulting in an unpredictable scaling. For this reason we felt that comparing amplitudes between conditions was not appropriate. However we are satisfied to look at differences between PC and MC for each condition. To summarize, we use the PC response as a standard for a direct thalamic input and much smaller responses in MC indicate the absence of a direct input.

6. *It would be helpful if justification for why the authors use X98-SST mice are used is presented when they are first mentioned (top of page 6).*

We now indicate that X98-SST mice were used to allow identification of Martinotti cells.

7. *Figure legend 1e: not sure what is meant by "separates of PCs"*

We have corrected this in the figure legend. "Plotting input-output (I-O) gain against threshold current separates PCs from MCs".

8. *Figure 4a: what is the difference between the traces on the left and right?*

The action potential frequencies differ. We now indicate in the figure legend: "10 Hz (left) and 30 Hz (right)".

9. *Figure 4g: what are the small red dots near the value 3? Are these individual data points (in black)?*

The red dashed line corresponds to the 3-fold increase in presynaptic frequency, from 10 Hz to 30 Hz. We clarified legends of figure 4g and 5g by adding "(red dashed line)" after "3-fold increase in presynaptic spike frequency". The black dots correspond to values from each of the n = 8 pairs.

10. *Page 4: should be "presubicular [recordings] in vivo"*

We replace "records" by "recordings".

11. *Page 5: not sure what is meant by "layer 1 as for other neocortical Martinotti cells"*

We now rephrase "layer 1 like Martinotti cells in somatosensory cortex (Ma et al. 2006)".

12. *Page 9: should be "in [recordings] from unidirectionally connected"*

We replaced "records" by "recordings".

Reviewer #4 (Remarks to the Author):

In the manuscripts titled, 'Activity dependent feedback inhibition supports head direction coding in the presubiculum' Simonnet and colleagues identify and characterize inputs to the pyramidal cell (PC) and somatostatin positive cells (SOM) of the presubiculum in mice slices. The manuscript presents numerous findings that center around the theme of identifying the circuit properties of the presubiculum. First, the authors show that between PC and SOM cells, presubicular inputs from the anterodorsal nucleus of the thalamus only synapse onto pyramidal cells. The authors demonstrate that the PC and SOM cells form a densely interconnected network including a high probability of reciprocal connections. The authors show

that the SOM-to-PC synapse efficiency has minimal activity dependence while the PC-to-SOM connection is highly activity dependent. The PC-to-SOM synapse efficacy increases with increasing PC activity, effectively unmuting the connection. The authors then compared reciprocally versus non-reciprocally connected SOM/ PC pairs for differences in inhibitory dynamics. The authors claim that their data demonstrates that feedback inhibition from a reciprocally connected SOM cell has minimal influence while non-reciprocally connected SOM cells have increased inhibitory influence on the PC. Finally, the authors present a computational model of the head direction circuit from which they conclude that the SOM cells support head direction coding in the presubiculum.

First and foremost, this is an outstanding manuscript both in terms of quality of research presented as well as in likely importance for the community. With regard to importance, the head direction circuit is a key component of the neural circuitry underlying the "brain's GPS" - a circuit celebrated by the 2014 Nobel Prize. The current work provides a vivid and much needed depiction of how the head direction signal is supported by the presubiculum, a key step in the relay of the head direction signal. This manuscript provides diverse, yet thematic, data regarding the functional structure of pyramidal and somatostatin positive interneurons circuitry in the presubiculum. This data answers open questions regarding how thalamic input is integrated by the presubiculum and makes substantial progress in building a complete picture of presubicular circuitry. With regard to the research, the questions posed are well formed and, in most cases, the empirical data provide conclusive answers to the questions.

I am highly enthusiastic about this manuscript and its potential to make a significant contribution to the field. However, there are several points that require attention prior to publication. I have described these as 'major points' below. Despite the label 'major' it should be noted that addressing the 'major points' need not take extensive work or time to address - I have indicated what I see to be the passable easy fix for each along with the not-so-easy fix. I have also included 'minor points' for the authors consideration for improved clarity / value of the manuscript.

Major points

Most critically, the model included in the manuscript strays too far from the current data. It misrepresents the data and introduces the distinct possibility of leading to false conclusions. By my understanding, the model architecture and synaptic dynamics both deviate from data reported in the manuscript in functionally consequential ways that prevent conclusions from being drawn from the model regarding the empirical data. With regard to synaptic dynamics, the model does not explicitly implement the reported activity dependent facilitation at PC to SOM cell synapses. Instead, it uses a winner take all mechanism which is not an acceptable approximation for the actual dynamics of the facilitating synapse. Additionally, the empirical data shows the latency of inhibitory feedback changes inhibitory efficacy but rather than allow the model to demonstrate this, the model design hard codes them into the simulation. Because they are hard coded, the model behavior cannot be used to draw a conclusion regarding the functional consequences of the latency data. With regard to architecture, a minor issue is that the connectivity ratios observed in the data are not literally replicated in the model and additional strong assumptions are made regarding how recurrent connections (called 'strong connections') fit into the architecture. On both fronts, there is no analysis as to whether these assumptions are required for the function of the model. Generally speaking, I'm in favor of including a model, especially because the data virtually beg for one. However, when I say this, I mean that the model should take replicating the observed data as its top priority and, from that standpoint, demonstrate the functional consequence of the observed phenomena. The current model uses abstractions of the current data (that I believe to be poor abstractions) which then weakens or eliminate the ability of the model to conclusively demonstrate that observed physiology has the functional properties cited by the authors. This is a significant flaw of the paper that must be addressed. Easy fix - drop the model, the paper does not need it to serve as a landmark work. Omitting the model may even spur a flurry of modeling work by others that will draw further attention to this data. Not-so-easy fix - redo the model to implement high fidelity realizations of the empirical data and from there demonstrate the

functional consequences.

We agree with the reviewer that some of the characteristics initially implemented in the model were not close enough to the experimental findings. We tried to overcome these limitations in a number of ways, following the reviewer's suggestions. First of all we modified the winner take all mechanism and we replaced it with a more "democratic" effect of presynaptic inputs in driving Martinotti cell activity. Now the activity of a single Martinotti cell is a function of the (weighted) sum of all its presynaptic inputs, while the feed-back inhibition is modified in proportion to the contribution of each one of the pre-synaptic pyramidal units. Therefore the more a pyramidal cell is dominant in determining the level of activation of a post-synaptic Martinotti cell, the more it will see the strength of the back-projection from that cell modified (methods, p. 21).

We also introduced facilitating dynamics at the PC to SOM synapse and we tested the effects of facilitating vs. depressing excitatory synaptic dynamics by parametrizing the strength of such modulation of synaptic efficacies and studying its effect on the stability of the directionally selective signal in the network. We show how, although depressing dynamics are not completely incompatible with coherent activity in the network, stable attractor dynamics require the presence of facilitation.

Concerning the more general critique about the level of abstraction present in the model, we note that the finality of this model is to provide a mechanistic test of how the interaction between a population of excitatory cells, and an inhibitory one, can give rise to continuous attractor dynamics suitable to represent and maintain directional information. The model also aims to describe the characteristics of directional tuning emerging in the two populations of cells: presence of directional tuning in the pyramidal population and the absence of directional selectivity among Martinotti interneurons. The model describes the network activity at the firing rate level and therefore it factors out finer temporal dynamics such as latencies, incorporating them in macroscopic elements such as connectivity strength. Our purpose here was to probe the possibility of continuous attractor dynamics when explicit PC to SOM interactions are included in the network. We agree that hard-wiring some aspects of the network dynamics (in our case by linking them to other model variables, for example using simultaneous firing rate as a measure for the likelihood of short spiking latency) is an abstraction of the data, however, we feel that the model still allows conclusions on the effects of their presence. Indeed we address the consequences of activity dependent variation in the feedback inhibition, while the particular structure of the initial connectivity is necessary for the emergence of selective activity in the PC layer and cannot be modified without compromising the formation of the activity bump.

The title, abstract, and discussion say that feedback inhibition supports head direction coding but the current data does not technically show this. It may be that the authors conclude this from the modeling work, however as noted above, I'm completely unconvinced by the model in its present form. The empirical data does not specifically show anything about head direction coding because it is done in slice. Easy fix: update the wording title, abstract, and discussion. My glowing enthusiasm of this manuscript is despite these statements not because of them. Dropping them will not decrease my excitement for seeing this manuscript in press. Not-so-easy fix: inactivate presubicular SOM cells in vivo while recording head direction cells, show that this reduces directional coding, and then conclude that local SOM cells support head direction coding.

We take the point that slice recordings cannot define the role of feedback inhibition in head direction coding *in vivo*. [redacted] To avoid overstatement here, we change the title of our ms to: "How activity dependent feedback inhibition may maintain head direction signals in mouse presubiculum". In the discussion, we now "suggest that [...] Martinotti cells become active during maintained directional signaling, and *could* support a form of working presubicular memory" (p. 14).

The analysis of recurrent versus non-recurrent inhibition does not conclusively demonstrate that there are functional differences in inhibitory dynamics between these two types of inhibition. Rather, the data show a difference in low-latency (< 10ms) IPSPs versus non-low-latency IPSPs (> 10ms). No data is shown regarding a difference in IPSP latency between recurrent and non-recurrent cell pairs. This data is needed to connect these functional differences resulting from latency to the connectivity differences. Such an analysis should compare the observed latency distributions to that expected by chance assuming that the PC cell has no influence on the SOM cell activity. This analysis would make it possible to identify the IPSP latencies that occur in recurrent pairs that occur more often than chance and, with this knowledge, consider the influence of IPSPs that arrive within this window. Similarly, this analysis could show if the distribution of IPSP latencies observed in non-recurrent pairs differs from what would be expected by chance.

Reviewer #4 points out that we have analyzed the effect of low-latency vs. longer latency inhibition to address the functional difference of recurrent vs. non-recurrent inhibition. The rationale for this is that recurrent PC-MC connectivity results in short latency feedback inhibition - a tight timing relation is the functional characteristic of a recurrent inhibitory connection.

We assume that non-recurrent MC activity is unrelated to PC spike timing. IPSPs would therefore occur at random timing with respect to the PC firing cycle. In a situation where the PC interspike interval is 50 ms (20 Hz firing), there is a 20% chance for an IPSP to occur in the first 10 ms after a PC spike. (This percentage would increase with the PC firing rate).

For recurrent connectivity, we now include new data to identify the timing of feed-back inhibition. We determine the delay for spike initiation in MCs cells triggered by a PC action potential in n = 4 PC-MC pairs (new panel in updated Fig. 8a): 84% of MC spikes have a delay of less than 8 ms after a PC spike (65 out of 77 spikes; n = 4 pairs). Considering a synaptic delay of 1-2 ms, the great majority of reciprocal IPSP latencies will be below 10 ms. In the case of reciprocal connections, the timing of the return IPSP therefore closely follows the PC spike, as it is illustrated in the existing Fig. 8a. The link between recurrent connectivity and the ensuing short latency feed-back IPSPs is now clearly stated in the text (p.9).

The authors report elsewhere (Nassar et al., 2015) that there is not a 1-to-1 mapping between the SOM label and Martinotti cells but the present work uses the name Martinotti cell for the fluorescently labeled SOM cells. Although the present paper notes that biocytin fills revealed Martinotti morphologies, no indication is given that analyses were restricted to those confirmed to have this morphology.

Most neurons presented in this ms were GFP positive cells of X98 mice. These cells are a homogeneous population, specific for Martinotti cells with axons ramifying in layer 1 (cf. Ma et al., 2006 for barrel cortex; Nassar et al., 2015 for presubiculum). Not only did we record exclusively from GFP+ neurons, but as described in the methods, we identified MC as low threshold spiking neurons of high input resistance and membrane potential in the range -65 to -50mV. Neurons which did not meet both criteria, highly reliable to identify Martinotti cells, were excluded.

For optogenetic experiments of Fig. 2 we had used SSTcre::tdTomato mice. Male homozygous SSTcre mice were crossed with female animals from the tdTomato reporter line (as now specified in methods), to obtain better specificity through paternal cre transmission. All recordings were made from layer 3, to select a more homogenous population, compared to our previous study, where we recorded from all layers of presubiculum.

Regarding stats:

Some indication that the included values in a statement like "gain was lower in PCs (0.373 {plus minus} 0.016 Hz.pA-1)" indicate mean and standard deviation should be given.

This is now specified (p.5 and elsewhere).

Additional attention should be given to indicating the n that goes into the included stats (e.g., n is not indicated in the first paragraph of the results or in the section titled 'Stable Martinotti cell inhibition. . .').

We now indicate n in the first paragraph of the results section (n = 80 MC and n = 87 PC) and in the section titled 'Stable Martinotti cell inhibition... ' (n = 8 pairs). A summary of all data points and n-values is given in Supplementary Tables 1 – 3.

Is n=4 sufficient to reliably estimate the decay rate of the PC-to-SOM synapse facilitation?

We have recorded more neuron pairs (now n = 7) to obtain a more reliable estimate of the decay of facilitation at the PC-to-SOM synapse. The text is modified accordingly “decay of synaptic efficacy and transfer back to baseline level, with a fast time constant of 0.74 and 0.97 seconds followed by a slower time constant of 12.96 and 18.94 s (n = 7, Fig. 6a-c)”. (p. 8).

Minor points

It would be powerful to include a putative mechanism for the facilitation. Is it presynaptic? Postsynaptic? Is it related to the PC to SOM activity dependent facilitation previously shown to be dependent on presynaptic NMDA receptors by Buchanan et al., (2012, Neuron)?

Presynaptic NMDA receptors control synaptic transmission at pyramidal cell inputs to Martinotti cells in layer 5 of visual cortex, and might also be involved in activity dependent facilitation in presubicular PC-to-MC synapses. We now include a reference to Buchanan et al. (2012) in the discussion (p.13).

A comparison of the number of reciprocal connections observed to the number of connections expected given the probability of SOM->PC and PC->SOM would be welcomed (I boarder on saying should be obligatory). This should include statistics indicating if it is significantly greater than expected by chance.

We add a sentence in the results, indicating that «28% of cell pairs (39 of 141) were reciprocally connected, a little more than the 22% expected given the probability for unilateral connections. For PCs that excited a MC, the probability of a reciprocal inhibitory connection was very high, 81% (39 out of 48 tested), while only 48% (39 out 80 tested) of MCs inhibiting a PC received reciprocal excitation». (p. 6).

Some indication of which subset of cell pairs were examined in subsequent analyses should be included (e.g., the one described in the section titled 'Stable Martinotti cell inhibition. . . ' or in section titled 'Repetitive stimulation unmutes. . .').

Usually pairs only served for one type of analysis and were not examined in multiple protocols (except basic properties and connectivity statistics).

The manuscript shows a 7-11s decay rate of the PC-to-SOM synapse - a discussion or, better yet, analysis of how this relates to the time scale of behavior would be welcomed. For example, what is the typical latency that an animal revisits a given head direction and what is the implication of this for the instantaneous distribution of how strongly facilitated the PC-to-SOM synapses are over the network? (This would be an exemplar analysis to do with the high fidelity variant of the model I argue for above.)

The decay rate of the enhancement of PC-to-MC synaptic transmission correlates well with the time scale of behavior (tens of seconds, p. 8). One could indeed estimate the distribution of the degree of synaptic facilitation present across the population of PC-to-MC synapses. We feel that this analysis probably goes beyond the current ms, but data from behaving animals should be addressed in future studies.

There is a jarring disconnect between the introduction and the first paragraph of the results. A Sentence or two at the outset of this section motivating hat these two cells types were of particular focus and that the

first step was to characterize their respective basic properties may help make this less disorienting.

We now add a sentence at the beginning of the results section, “In order to elucidate the functional role of Martinotti cells in the presubicular microcircuit, we first characterized the basic properties of Martinotti cells (MC) and pyramidal cells (PC) in superficial layer 3 of mouse presubiculum. In total, data from 166 PCs and 161 MC recorded in horizontal slices (Fig. 1) from 60 animals are included in this study.” (p. 5)

The sudden appearance of PV+ cell analysis in the section titled 'How in vivo head direction signaling. . .' suddenly made me wonder why they had been ignored otherwise. Some improved introduction may mitigate this. In the discussion, I would welcome some time spent on incorporating how the PV+ cells might fit into all that has been shown in the paper.

The reviewer is surely right that role of PV neurons in the presubicular microcircuit deserves further investigation. While this work focuses on Martinotti cells, we plan next to examine the recruitment of fast-spiking PV interneurons. In the present discussion we note that “PV neurons with depressing synapses (Supplementary Fig. 4) are not part of the attractor. We suggest that during fast head turns, when Martinotti cells are not recruited, the system may switch to a relay type function.” We now add a reference to Preston-Ferrer et al. (2016) indicating “Presubicular fast-spiking interneurons fire at higher rates during rotation¹⁰, when the population of active head direction cells shifts quickly. The excitatory inputs received by PV neurons continuously changes to different sets of synapses, and for each transient head direction, PV neurons will rapidly provide inhibition with depressing dynamics.” (p. 14).

Author response image 1

[redacted]

REVIEWERS' COMMENTS:

Reviewer #1 (Remarks to the Author):

The authors addressed all the comments. The manuscript deserve publication.

Reviewer #2 (Remarks to the Author):

The authors have significantly improved the manuscript upon this revision.

However, I would like to point out that one of my major concerns (major point 1) related to the artificial experimental conditions of studying the effect of dendritic inhibition on somatic spike generation -- namely the use of artificially enhanced hyperpolarizing inhibition (i.e. low chloride intracellular solution), was acknowledged in the rebuttal, but this concern was not experimentally addressed or alleviated.

The authors' proposal that the dominant effect of a major dendritic inhibitory cell type on synaptic integration would occur through some intricate changes of somatic action potential AHP and peak (Fig. 8) remains difficult to comprehend. In my view, more solid conclusions would have required experimental conditions under which dendritic integration of excitatory and inhibitory inputs had been properly mimicked.

Reviewer #3 (Remarks to the Author):

The authors have address a number of my concerns. However, I do still have a few comments that should be addressed.

1. I remain unsure about the range of volumes of virus injected (50-150 nl). In the author's rebuttal, they seem to imply that all five mice recieved 150nl? But then mention even larger injection volumes (300ml)? Were the patterns of expression the same for this range? When were the smaller ranges used (> 150 nl)? Please clarify these points. In addition, I do still think an image of ATN (and surrounding structures) to illustrate the extent of virus expression (instead of just a table) would strengthen the paper.
2. The change on page 17 with reference to the use of different species does not discuss the caveats necessitated by the differences in network microstructure between them.
3. I think the expanded and clarified figures/legends and the response of the authors is an improvement, but it's still not clear why no direct comparison between MC cells in control and TTX are made? It's very clear that MC and PC are distinguishable, but is there a difference between MC in control and TTX?

Reviewer #4 (Remarks to the Author):

My initial opinion of the manuscript was very high and I had only a few major concerns. The authors have been extremely diligent in addressing my concerns proactively and thoroughly. Of course there will always be the "I would still prefer to see it this other way . . ." but I have no concerns about either the integrity, validity, or reliability of the work presented here. As I emphasized in my initial review, I firmly believe that this will be a highly impactful paper of broad appeal.

Point-by-point response to REVIEWERS' COMMENTS:

Reviewer #1 (Remarks to the Author):

The authors addressed all the comments. The manuscript deserve publication.

Reviewer #2 (Remarks to the Author):

The authors have significantly improved the manuscript upon this revision. However, I would like to point out that one of my major concerns (major point 1) related to the artificial experimental conditions of studying the effect of dendritic inhibition on somatic spike generation -- namely the use of artificially enhanced hyperpolarizing inhibition (i.e. low chloride intracellular solution), was acknowledged in the rebuttal, but this concern was not experimentally addressed or alleviated.

Page 12, last two lines, now reads :

« Our pipette solution was designed to artificially enhance the driving force for chloride, by using low-chloride in the internal solution, increasing our ability to detect inhibitory synaptic events and to distinguish them from failures. »

The authors' proposal that the dominant effect of a major dendritic inhibitory cell type on synaptic integration would occur through some intricate changes of somatic action potential AHP and peak (Fig. 8) remains difficult to comprehend. In my view, more solid conclusions would have required experimental conditions under which dendritic integration of excitatory and inhibitory inputs had been properly mimicked.

We insert a sentence, p. 14,

« Further work is needed to experimentally test how, in presubiculum, inhibitory and excitatory inputs impinging on postsynaptic PC dendrites are integrated. »

Reviewer #3 (Remarks to the Author):

The authors have address a number of my concerns. However, I do still have a few comments that should be addressed.

1. I remain unsure about the range of volumes of virus injected (50-150 nl). In the author's rebuttal, they seem to imply that all five mice recieved 150nl? But then mention even larger injection volumes (300ml)? Were the patterns of expression the same for this range? When were the smaller ranges used (> 150 nl)? Please clarify these points.

As stated in the ms, page 15 : "The injected volume was 150 nl, to be as specific as possible (cf. Supplementary Fig. 1), but with enough spread to cover ATN."

In addition, I do still think an image of ATN (and surrounding structures) to illustrate the extent of virus expression (instead of just a table) would strengthen the paper.

Two new images of the thalamic injection site have been included in Supplementary Figure 2.

2. The change on page 17 with reference to the use of different species does not discuss the caveats necessitated by the differences in network microstructure between them.

We now acknowledge inter-species differences in network microarchitecture, page 16 : "Even though inter-species differences in network microstructure may exist, HD direction cells have been described in presubiculum in mice and in rats^{3,4,10,29}, and intrinsic neuronal properties and projection targets are similar in both species^{15,34,64}."

3. I think the expanded and clarified figures/legends and the response of the authors is an improvement, but it's still not clear why no direct comparison between MC cells in control and TTX are made? It's very clear that MC and PC are distinguishable, but is there a difference between MC in control and TTX?

Yes. We now state in the figure legend for Figure 2 : "For PCs, the charge transfer was significantly reduced (**** Wilcoxon signed rank test $P < 0.0001$) but still present in TTX/4AP condition. For MC cells, the measured charge transfer was also significantly reduced (** Wilcoxon signed rank test $P=0.0078$) to values close to zero in TTX/4AP condition".

Reviewer #4 (Remarks to the Author):

My initial opinion of the manuscript was very high and I had only a few major concerns. The authors have been extremely diligent in addressing my concerns proactively and thoroughly. Of course there will always be the "I would still prefer to see it this other way . . ." but I have no concerns about either the integrity, validity, or reliability of the work presented here. As I emphasized in my initial review, I firmly believe that this will be a highly impactful paper of broad appeal.